# Advancing Training Efficiency of Deep Spiking Neural Networks through Rate-based Backpropagation

**Chengting Yu**[1,2], **Lei Liu**[2], **Gaoang Wang**[2], **Erping Li**[1,2], **Aili Wang**[1,2*]
[1] College of Information Science and Electronic Engineering, Zhejiang University
[2] ZJU-UIUC Institute, Zhejiang University
chengting.21@intl.zju.edu.cn, ailiwang@intl.zju.edu.cn

## Abstract

Recent insights have revealed that rate-coding is a primary form of information representation captured by surrogate-gradient-based Backpropagation Through Time (BPTT) in training deep Spiking Neural Networks (SNNs). Motivated by these findings, we propose rate-based backpropagation, a training strategy specifically designed to exploit rate-based representations to reduce the complexity of BPTT. Our method minimizes reliance on detailed temporal derivatives by focusing on averaged dynamics, streamlining the computational graph to reduce memory and computational demands of SNNs training. We substantiate the rationality of the gradient approximation between BPTT and the proposed method through both theoretical analysis and empirical observations. Comprehensive experiments on CIFAR-10, CIFAR-100, ImageNet, and CIFAR10-DVS validate that our method achieves comparable performance to BPTT counterparts, and surpasses state-of-the-art efficient training techniques. By leveraging the inherent benefits of rate-coding, this work sets the stage for more scalable and efficient SNNs training within resource-constrained environments. Our code is available at https://github.com/Tab-ct/rate-based-backpropagation.

## 1  Introduction

Spiking Neural Networks (SNNs) are conceptualized as biologically inspired neural systems, incorporating spiking neurons that closely mimic biological neural dynamics [46, 56]. Unlike Artificial Neural Networks (ANNs) based on continuous data representations, SNNs adopt spike-coding strategies to facilitate data transmission through discrete binary spike trains [52]. The intrinsic binary mechanism eliminates the need for the extensive multiply-accumulate operations typically required for synaptic connectivity [56], thereby enhancing energy efficiency and inference speed when deployed on neuromorphic hardware systems [1, 10, 54].

The mainstream training methods for SNNs primarily utilize Backpropagation Through Time (BPTT) with surrogate gradients to overcome non-differentiable spike events, allowing SNNs to achieve comparable results with ANNs counterparts [51, 62, 74]. However, the direct training method necessitates the storage of all temporal activations for backward propagation across the network's depth and duration, leading to high training costs in terms of both computational time and memory demands [43, 86, 35, 77, 76, 47, 13]. To alleviate memory burdens, online training techniques have been developed that partially decouple the time dependencies of backward computations in BPTT [2, 4, 76, 48, 89]. However, online methods still require iterative computations based on the time dimension, increasing training time complexity as the number of timesteps grows.

---

*Corresponding author.

Observed across most biological sensory systems, rate coding is a phenomenon where information is encoded through the rate of neuronal spikes, regardless of precise spike timing [52, 64, 23]. Recent explorations into spike representation have demonstrated the significant role of rate coding in enhancing the robustness of SNNs, further confirming its dominant position as the encoding representation in networks [38, 60, 18]. A significant observation has shown that BPTT-trained SNNs on static benchmark exhibit spike representation primarily following the rate-coding manner by highlighting strong similarities in representation between SNNs and their ANN counterparts [44]. A similar conclusion resonated with findings in fields of adversarial attacks, where recent methods significantly benefit from rate-based representations to enhance attack effectiveness [6, 29, 50].

Motivated by rate coding's status as the most effective and predominant form of representation in SNNs, we posit that targeted training based on rate-based information could offer a high cost-effectiveness ratio. We propose to decouple BPTT based on rate-coding approximation and simplify rate-based derivative computations to a single spatial backpropagation. We further provide theoretical analysis and empirical evidence to reveal the rationality of the gradient approximation between BPTT and the proposed method. Experimental results demonstrate that the proposed method achieves performance comparable to BPTT counterparts while significantly reducing memory and computational demands. Comparison results also indicate that the proposed method outperforms state-of-the-art efficient training methods on benchmarks. We expect our work to facilitate more efficient and scalable training for SNNs in resource-constrained environments. Our main contributions are as follows:

- We propose rate-based backpropagation that leverages rate-coded information for efficient training of deep SNNs. This method simplifies the computational graph by decoupling and compressing temporal dependencies, reducing training time and memory requirements.
- Alongside the proposed method, we conduct theoretical analysis and empirical validation to demonstrate its effectiveness in approximating the gradient computations performed by BPTT-based SNNs training.
- We conduct experiments on CIFAR-10, CIFAR-100, CIFAR10-DVS, and ImageNet, demonstrating that our proposed method matches the comparable performance of the BPTT counterpart and achieves state-of-the-art results among efficient SNN training methods.

## 2   Related Work

**Training Methods for Deep SNNs.** Deep SNNs are trained primarily through two principal strategies: (1) conversion methods that establish links between SNNs and ANNs through equivalent closed-form mappings, and (2) direct training from scratch utilizing Backpropagation Through Time (BPTT). Conversion methods develop closed-form formulations for spike representations [39, 67, 72, 88, 73, 47], enabling seamless transitions of pre-trained ANNs into SNNs and facilitating comparable performance on large-scale datasets [8, 15, 27, 59, 57, 12, 42, 16]. However, the precision of these mappings under ultra-low latency conditions is not consistently reliable, often necessitating extensive time steps to accumulate spikes, which may compromise performance [7, 40, 31, 28, 34]. Direct training methods permit SNNs' performance with extremely low time steps by employing BPTT along with surrogate gradients to compute derivatives of discrete spiking events [51, 62, 74, 22, 81, 87, 85, 43, 66, 69, 13]. The strategy fosters innovation in SNN-specific modules, including optimized neurons, synapses, and network architectures, thereby enhancing performance [25, 21, 20, 17, 79, 83, 24, 80, 61]. Despite the advantages of low latency, direct training imposes substantial memory and time burdens to maintain the backward computational graph [43, 86, 35, 77, 76, 47, 13]. To mitigate training costs associated with direct methods, light training strategies have attracted considerable attention [49, 35, 86, 55, 70]. Several studies have explored the concept of decoupling the forward and backward passes in SNNs, which generally assumes that neuronal dynamics follow deterministic processes and aims to establish closed-form fixed-point equivalences between spike representations and corresponding rate-based activations [72, 73, 77, 47, 68]. Drawing on online training techniques from recurrent neural networks, several studies have adapted the principles of Real-time Recurrent Learning (RTRL) [71] to streamline the online training process for SNNs, aiming to decrease memory demands while preserving biologically plausible online properties of the networks [84, 2, 4, 82, 55, 76, 47, 89]. The online methodologies have proven effective in large-scale tasks [76, 47, 89]. Nevertheless, the significant time costs associated with training methods continue to challenge SNNs' broader application.

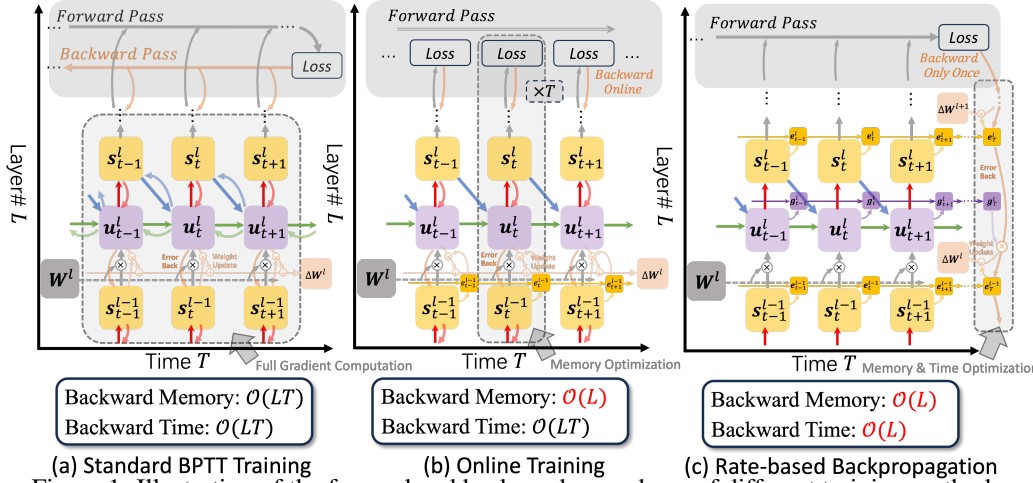

Figure 1: Illustration of the forward and backward procedures of different training methods.

**Spike Coding in SNNs.** SNNs transmit information through spike trains [52], with encoding mechanisms classified into temporal and rate coding. Temporal coding is defined on firing times, employed by several direct trainings [49, 75, 88] and ANN-to-SNN conversions [26, 65], is noted for its low energy consumption due to sparse spiking. However, temporal coding schemes often require specialized neuron configurations and are generally effective only on simpler datasets [26, 65, 88]. Conversely, rate coding is widely adopted across both conversion [12, 15, 16, 27, 36, 57, 59, 78] and direct training approaches [73, 77, 47], consistently achieving superior performance and facilitating low-latency operations [77, 47]. Moreover, rate coding has demonstrated significant potential in enhancing the robustness of SNNs against adversarial attacks [38, 60, 18], with attack methods specifically designed to exploit rate-based representations showing promise in surpassing benchmarks for SNNs defense against attacks [6, 30, 50]. By employing representation similarity analysis to compare BPTT-trained SNNs with their ANN counterparts, Li et al. [44] has indicated that rate coding serves as the primary mode of information representation [44]. Inspired by previous findings, we consider that rate-coded information represents the most effective and predominant form of signal expression in SNNs, and the targeted training based on rate-based spike representations may offer a high cost-effectiveness ratio. Therefore, we propose to decouple BPTT towards rate-based backpropagation with the purpose of enhancing the efficiency of SNNs training.

## 3 Preliminaries

### 3.1 Spiking Neural Networks

Inspired by the brain's ability to transmit information through discrete spikes, the Leaky Integrate-and-Fire (LIF) model serves as the basic building block of SNNs due to its simplicity. For practical implementation of SNNs based on connected spiking neurons, the dynamics of the LIF model are typically rendered in a discrete iterative format:

$$\boldsymbol{u}_t^l = \lambda(\boldsymbol{u}_{t-1}^l - V_{\text{th}}\boldsymbol{s}_{t-1}^l) + \boldsymbol{W}^l \boldsymbol{s}_t^{l-1}, \quad \boldsymbol{s}_t^l = H(\boldsymbol{u}_t^l - V_{\text{th}}) \tag{1}$$

where $\boldsymbol{u}_t^l$ and $\boldsymbol{s}_t^l$ represent the membrane potential and output spike of neurons in layer $l$ at time $t$, respectively. $\boldsymbol{W}^l$ denotes the linear synaptic connections between layers $l-1$ and $l$, and $\lambda$ acts as the decay term for the membrane potential. The Heaviside step function, $H(\cdot)$, determines spike generation, ensuring $\boldsymbol{s}_t^l$ in binary forms. Noting that $H(\cdot)$ is not differentiable, SNNs' direct training employs surrogate gradients to achieve error propagation by creating various pseudo-derivatives [51, 74, 19], following the basic idea of Straight-Through Estimator (STE) [3].

### 3.2 Training SNNs with BPTT

The network outputs at each timestep $t$ are given by $\boldsymbol{o}_t = \boldsymbol{W}^L \boldsymbol{s}_t^L$, where $\boldsymbol{W}^L$ denotes the classifier's weights. Classification is based on the average of these outputs across all timesteps, computed as $\boldsymbol{y}_{\text{pred}} = \frac{1}{T}\sum_{t=1}^{T}\boldsymbol{o}_t$. The loss function $\mathcal{L}$ is defined over averaged outputs and is typically formulated as $\mathcal{L} = \ell\left(\frac{1}{T}\sum_{t=1}^{T}\boldsymbol{o}_t, \boldsymbol{y}\right)$, where $\boldsymbol{y}$ represents the true labels and $\ell$ could be the cross-entropy function, as noted in various studies [87, 48, 19, 69]. BPTT unfolds the iterations described in Eq. (1), and propagates gradients back along the computational graphs across both temporal and spatial dimensions, as illustrated in Fig. 1a. The gradients of the membrane potential $\boldsymbol{u}$ incorporate elements

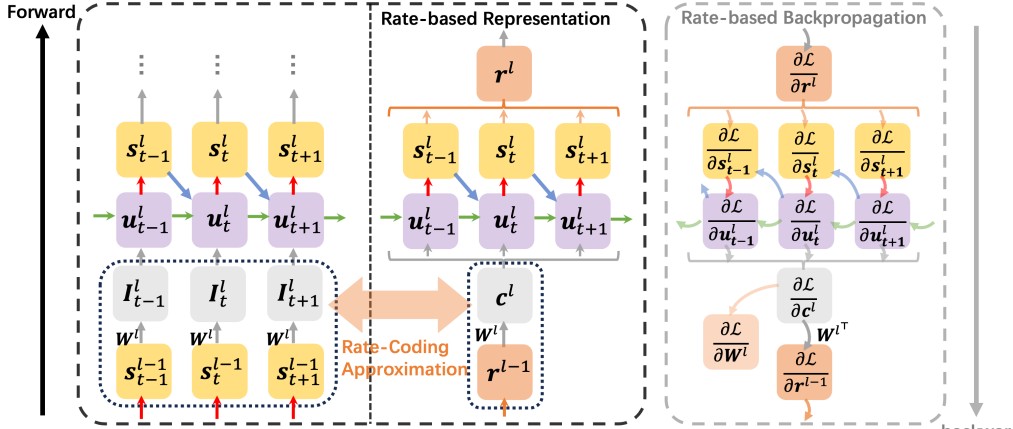

Figure 2: The implementation of rate-based backpropagation across layers. A rate-coding approximation is utilized for the forward procedure to connect average inputs with rate outputs, enabling fast rate-based error backpropagation throughout the training process.

from both (spatial) spike generation and (temporal) potential accumulation, expressed as:

$$\frac{\partial \mathcal{L}}{\partial \boldsymbol{u}_t^l} = \frac{\partial \mathcal{L}}{\partial \boldsymbol{s}_t^l} \frac{\partial \boldsymbol{s}_t^l}{\partial \boldsymbol{u}_t^l} + \frac{\partial \mathcal{L}}{\partial \boldsymbol{u}_{t+1}^l} \left( \frac{\partial \boldsymbol{u}_{t+1}^l}{\partial \boldsymbol{u}_t^l} + \frac{\partial \boldsymbol{u}_{t+1}^l}{\partial \boldsymbol{s}_t^l} \frac{\partial \boldsymbol{s}_t^l}{\partial \boldsymbol{u}_t^l} \right)$$

$$= \frac{\partial \mathcal{L}}{\partial \boldsymbol{s}_t^l} \frac{\partial \boldsymbol{s}_t^l}{\partial \boldsymbol{u}_t^l} + \sum_{\tau > t} \frac{\partial \mathcal{L}}{\partial \boldsymbol{s}_\tau^l} \frac{\partial \boldsymbol{s}_\tau^l}{\partial \boldsymbol{u}_\tau^l} \prod_{i=\tau-1}^{t} \left( \frac{\partial \boldsymbol{u}_{i+1}^l}{\partial \boldsymbol{u}_i^l} + \frac{\partial \boldsymbol{u}_{i+1}^l}{\partial \boldsymbol{s}_i^l} \frac{\partial \boldsymbol{s}_i^l}{\partial \boldsymbol{u}_i^l} \right)$$

(2)

Subsequently, the weight update for layer $l$ is determined among all timesteps $T$, i.e. $\nabla_{\boldsymbol{W}^l} \mathcal{L} = \sum_{t=1}^{T} \frac{\partial \mathcal{L}}{\partial \boldsymbol{u}_t^l} \frac{\partial \boldsymbol{u}_t^l}{\partial \boldsymbol{W}^l} = \sum_{t=1}^{T} \frac{\partial \mathcal{L}}{\partial \boldsymbol{u}_t^l} \boldsymbol{s}_t^{l-1\top}$, and the gradient is further propagated to previous layers through the linear part by $\frac{\partial \mathcal{L}}{\partial \boldsymbol{s}_t^{l-1}} = \frac{\partial \mathcal{L}}{\partial \boldsymbol{u}_t^l} \boldsymbol{W}^{l\top}$.

## 4 Rate-based Backpropagation for SNNs Training

### 4.1 Derivation of Rate-based Backpropagation

**Incorporating rate-based representation.** Under the rate coding assumption, essential information is effectively encapsulated within the spike frequency averages. We start by defining the rate-based representation as an approximation for the forward procedure in SNNs, as shown in Figure 2. The average firing rate at each layer $l$, denoted as $\boldsymbol{r}^l$, is calculated as the expected value of the spike outputs $\boldsymbol{s}_t^l$ over the temporal dimension $\boldsymbol{r}^l = \mathbb{E}[\boldsymbol{s}_t^l] = \frac{1}{T} \sum_{t \leq T} \boldsymbol{s}_t^l$.

Considering the forward propagation through linear operators with weights $\boldsymbol{W}^l$ that compute the inputs as $\boldsymbol{I}_t^l = \boldsymbol{W}^l \boldsymbol{s}_t^{l-1}$, instead of transmitting distinct spikes over multiple timesteps, we transform the average rates into average inputs $\boldsymbol{c}^l$ in the approximate representation:

$$\boldsymbol{c}^l = \mathbb{E}[\boldsymbol{I}_t^l] = \mathbb{E}[\boldsymbol{W}^l \boldsymbol{s}_t^{l-1}] = \boldsymbol{W}^l \mathbb{E}[\boldsymbol{s}_t^{l-1}] = \boldsymbol{W}^l \boldsymbol{r}^{l-1}.$$

Supposing input representations are well captured within $\boldsymbol{c}^l$, we approximate the exact inputs with the average inputs for all timesteps, $\boldsymbol{I}_t^l \approx \boldsymbol{c}^l$, and follow the neuronal dynamics in Eq. (1) to derive the output rates $\boldsymbol{r}^l = \mathbb{E}[\boldsymbol{s}_t^l]$. With the rate-coding approximation in place, we can derive the gradients with respect to the weights in the linear part based on the error propagated through the average inputs:

$$\left( \nabla_{\boldsymbol{W}^l} \mathcal{L} \right)_{\text{rate}} \equiv \frac{\partial \mathcal{L}}{\partial \boldsymbol{c}^l} \frac{\partial \boldsymbol{c}^l}{\partial \boldsymbol{W}^l} = \frac{\partial \mathcal{L}}{\partial \boldsymbol{c}^l} \boldsymbol{r}^{l-1\top}$$

(3)

**Handling temporal dependency during backward.** For back-propagating the error, the linear parts operate smoothly as $\frac{\partial \boldsymbol{c}^l}{\partial \boldsymbol{r}^{l-1}} = \boldsymbol{W}^{l\top}$. The next step is to define the correlation between the averages of inputs and output spike rates, $\frac{\partial \boldsymbol{r}^l}{\partial \boldsymbol{c}^l}$, within the neurons of layer $l$. Since there is no deterministic relationship between $\boldsymbol{r}^l$ and $\boldsymbol{c}^l$, we first look into the influence of separated inputs following the exact

gradients in Eq. (2):

$$\frac{\partial \boldsymbol{s}_\tau^l}{\partial \boldsymbol{I}_t^l} = \frac{\partial \boldsymbol{s}_\tau^l}{\partial \boldsymbol{u}_t^l} = \begin{cases} \frac{\partial \boldsymbol{s}_\tau^l}{\partial \boldsymbol{u}_\tau^l} \prod_{i=\tau-1}^t \left( \frac{\partial \boldsymbol{u}_{i+1}^l}{\partial \boldsymbol{u}_i^l} + \frac{\partial \boldsymbol{u}_{i+1}^l}{\partial \boldsymbol{s}_i^l} \frac{\partial \boldsymbol{s}_i^l}{\partial \boldsymbol{u}_i^l} \right) & \text{if } \tau \geq t \\ \frac{\partial \boldsymbol{s}_t^l}{\partial \boldsymbol{u}_t^l} & \text{if } \tau = t \\ 0 & \text{if } \tau < t \end{cases} \tag{4}$$

By accumulating the intricate dynamics over time, we can derive the gradients of the overall spikes with respect to the inputs at time $t$:

$$\boldsymbol{\varkappa}_t^l = \sum_\tau \frac{\partial \boldsymbol{s}_\tau^l}{\partial \boldsymbol{I}_t^l} = \left( \frac{\partial \boldsymbol{s}_t^l}{\partial \boldsymbol{u}_t^l} + \sum_{\tau > t} \frac{\partial \boldsymbol{s}_\tau^l}{\partial \boldsymbol{u}_\tau^l} \prod_{i=\tau-1}^t \left( \frac{\partial \boldsymbol{u}_{i+1}^l}{\partial \boldsymbol{u}_i^l} + \frac{\partial \boldsymbol{u}_{i+1}^l}{\partial \boldsymbol{s}_i^l} \frac{\partial \boldsymbol{s}_i^l}{\partial \boldsymbol{u}_i^l} \right) \right) \tag{5}$$

Here, with rate-coding approximating $\boldsymbol{I}_t^l \approx \boldsymbol{c}^l$, we follow the idea of Straight-Through Estimator [3] and define the backward gradients as $\frac{\partial \boldsymbol{I}_t^l}{\partial \boldsymbol{c}^l} = Id$, with $Id$ representing the identity matrix. Then, we can derive the surrogate gradients of neural dynamics through the mean estimator:

$$\left( \frac{\partial \boldsymbol{r}^l}{\partial \boldsymbol{c}^l} \right)_{\text{rate}} \equiv \sum_\tau \left( \frac{\partial (\mathbb{E}\left[\boldsymbol{s}_t^l\right])}{\partial \boldsymbol{I}_\tau^l} \frac{\partial \boldsymbol{I}_\tau^l}{\partial \boldsymbol{c}^l} \right) = \frac{1}{T} \sum_t \sum_\tau \left( \frac{\partial \boldsymbol{s}_t^l}{\partial \boldsymbol{I}_\tau^l} \right) = \mathbb{E}\left[ \boldsymbol{\varkappa}_t^l \right] \tag{6}$$

With the compressed gradients of neuron parts, the error backpropagation of the rate-based representation is then determined, dependent only on the spatial domain:

$$\left( \frac{\partial \mathcal{L}}{\partial \boldsymbol{c}^l} \right)_{\text{rate}} = \left( \frac{\partial \mathcal{L}}{\partial \boldsymbol{c}^L} \prod_{i=L-1}^l \left( \frac{\partial \boldsymbol{c}^{i+1}}{\partial \boldsymbol{r}^i} \left( \frac{\partial \boldsymbol{r}^i}{\partial \boldsymbol{c}^i} \right)_{\text{rate}} \right) \right) = \left( \frac{\partial \mathcal{L}}{\partial \boldsymbol{c}^L} \prod_{i=L-1}^l \left( \boldsymbol{W}^{i\top} \mathbb{E}\left[ \boldsymbol{\varkappa}_t^l \right] \right) \right) \tag{7}$$

where we define the objective $\mathcal{L} = \frac{1}{T}\ell(\mathbb{E}[\boldsymbol{o}_t], y) = \frac{1}{T}\ell(\boldsymbol{c}^L, \boldsymbol{y})$. Note that the rate-based representation, while instrumental in constructing the backward computational graph for learning, does not necessitate actual implementation during the forward pass.

## 4.2 Rate-based Gradient Computation for Memory and Time Efficiency

As previously discussed, rate-based backpropagation can be executed on spatial-dimension computation by decoupling BPTT. We now show how rate-based backpropagation can be efficiently implemented within the overall learning framework. As depicted in Figure 2b, online schemes apply eligibility traces $\boldsymbol{e}_t^l$ locally within neurons to store historical information, effectively blocking backward access to past gradients. The gradient computation is optimized by compressing all past temporal dependencies into $\boldsymbol{e}_t^l$. Similarly, we utilize iterative variables $\{\boldsymbol{g}_t^l\}_{l \leq L}$ and $\{\boldsymbol{e}_t^l\}_{l \leq L}$ as the accumulated post- and pre-synaptic dependencies, synchronously recorded during the neural dynamics computations. The iteration of $\{\boldsymbol{e}_t^l\}_{l \leq L}$ dynamically records the firing rates, where $\boldsymbol{e}_t^l = \frac{1}{t}((t-1)\boldsymbol{e}_{t-1}^l + \boldsymbol{s}_t^l)$, and it is straightforward to derive $\boldsymbol{r}^l = \boldsymbol{e}_T^l$. Considering the surrogate gradients of neural dynamics, $\frac{\partial \boldsymbol{r}^l}{\partial \boldsymbol{c}^l}$, to estimate future-dependent terms outlined in Eq. 5, we first construct equivalent eligibility trace forms, $\{\boldsymbol{\rho}_t^l\}_{t \leq T}$, with iterative expressions starting at $\boldsymbol{\rho}_1^l = 1$:

$$\boldsymbol{\rho}_t^l = 1 + \boldsymbol{\rho}_{t-1}^l \left( \frac{\partial \boldsymbol{u}_t^l}{\partial \boldsymbol{u}_{t-1}^l} + \frac{\partial \boldsymbol{u}_t^l}{\partial \boldsymbol{s}_{t-1}^l} \frac{\partial \boldsymbol{s}_{t-1}^l}{\partial \boldsymbol{u}_{t-1}^l} \right) = 1 + \sum_{\tau < t} \prod_{i=t-1}^\tau \left( \frac{\partial \boldsymbol{u}_{i+1}^l}{\partial \boldsymbol{u}_i^l} + \frac{\partial \boldsymbol{u}_{i+1}^l}{\partial \boldsymbol{s}_i^l} \frac{\partial \boldsymbol{s}_i^l}{\partial \boldsymbol{u}_i^l} \right) \tag{8}$$

with the equivalence that:

$$\begin{aligned} \sum_t \boldsymbol{\varkappa}_t^l &= \sum_t \left( \frac{\partial \boldsymbol{s}_t^l}{\partial \boldsymbol{u}_t^l} + \sum_{\tau > t} \left( \frac{\partial \boldsymbol{s}_\tau^l}{\partial \boldsymbol{u}_\tau^l} \prod_{i=\tau-1}^t \left( \frac{\partial \boldsymbol{u}_{i+1}^l}{\partial \boldsymbol{u}_i^l} + \frac{\partial \boldsymbol{u}_{i+1}^l}{\partial \boldsymbol{s}_i^l} \frac{\partial \boldsymbol{s}_i^l}{\partial \boldsymbol{u}_i^l} \right) \right) \right) \\ &= \sum_t \left( \frac{\partial \boldsymbol{s}_t^l}{\partial \boldsymbol{u}_t^l} \left( 1 + \sum_{\tau < t} \prod_{i=t-1}^\tau \left( \frac{\partial \boldsymbol{u}_{i+1}^l}{\partial \boldsymbol{u}_i^l} + \frac{\partial \boldsymbol{u}_{i+1}^l}{\partial \boldsymbol{s}_i^l} \frac{\partial \boldsymbol{s}_i^l}{\partial \boldsymbol{u}_i^l} \right) \right) \right) = \sum_t \left( \frac{\partial \boldsymbol{s}_t^l}{\partial \boldsymbol{u}_t^l} \boldsymbol{\rho}_t^l \right) \end{aligned} \tag{9}$$

By iteratively accumulating $\boldsymbol{g}_t^l = \frac{1}{t}((t-1)\boldsymbol{g}_{t-1}^l + \frac{\partial \boldsymbol{s}_t^l}{\partial \boldsymbol{u}_t^l}\boldsymbol{\rho}_t)$, we obtain $\boldsymbol{g}_T^l = \mathbb{E}[\frac{\partial \boldsymbol{s}_t^l}{\partial \boldsymbol{u}_t^l}\boldsymbol{\rho}_t^l] = \mathbb{E}[\boldsymbol{\varkappa}_t^l]$. Now, we have collapsed the required computation graph through the iterative calculation to complexity $\mathcal{O}(L)$. The rate-based propagation is then conducted in one go, relying only on the intermediate variables $\boldsymbol{e}_T^l$, $\boldsymbol{g}_T^l$, and $\boldsymbol{W}^l$, within one-time spatial-dimension backpropagation.

## 4.3 Connecting Error Backward of Rate-based Backpropagation to BPTT

Having derived the fundamental form of rate-based backpropagation through the rate-encoding approximation, we now explore potential divergences with BPTT during error propagation. Although rate-based backpropagation is derived from the approximated forward pass, it still provides valid gradients for the original network parameters.

The primary divergence between rate-back and BPTT in backward computation primarily arises from the assumptions regarding the approximation of rate-based representation through mean estimators, as outlined in Eq.(3) and Eq.(6). The rate-coding motivations establish equivalence with BPTT by assuming temporal components are independent, which is formalized in Theorem 1.

**Theorem 1.** *Given $\delta_t^{(s^l)} = \frac{\partial \mathcal{L}}{\partial s_t^l}$ that refers to gradients computed following the chain rule of BPTT in Eq. (2), and $\kappa_t^l = \sum_\tau \frac{\partial s_t^l}{\partial I_\tau^l}$ (where $\mathbb{E}\left[\kappa_t^l\right] = \mathbb{E}\left[\varkappa_t^l\right]$ in Eq.(6-7)), if $\mathbb{E}\left[\delta_t^{(s^l)}\kappa_t^l\right] = \mathbb{E}\left[\delta_t^{(s^l)}\right]\mathbb{E}\left[\kappa_t^l\right]$ holds for $\forall l$, we have $\mathbb{E}\left[\delta_t^{(s^l)}\right] = \left(\frac{\partial \mathcal{L}}{\partial r^l}\right)_{rate}$. Furthermore, given $\delta_t^{(I^l)} = \frac{\partial \mathcal{L}}{\partial I_t^l}$, if $\mathbb{E}\left[\delta_t^{(I^l)}s_t^{l-1}\right] = \mathbb{E}\left[\delta_t^{(I^l)}\right]\mathbb{E}[s_t^{l-1}]$ for $\forall l$, we then obtain $(\nabla_{W^l}\mathcal{L})_{rate} = \frac{1}{T}(\nabla_{W^l}\mathcal{L})$. Here, $\mathbb{E}\left[x_t\right] = \frac{1}{T}\sum_t x_t$ refers the mean value of tensor $x_t$ over temporal dimension $T$.*

To confirm our hypotheses, we carried out empirical experiments, the results of which are detailed in the experimental section. Our empirical findings support the core assumptions outlined in Theorem 1, demonstrating the relative independence between $\delta_t^{(s^l)}$ and $\kappa_t^l$ (Figure 3a,b), as well as between $\delta_t^{(I^l)}$ and $s_t^l$ (Figure 3c). For minor discrepancies that may arise, we introduced Theorem 2, which tolerates small deviations and confirms that approximation errors in rate-based backpropagation can be effectively bounded, ensuring the robustness of training under practical conditions.

**Theorem 2.** *For gradients $\delta_t^{(s^l)} = \frac{\partial \mathcal{L}}{\partial s_t^l}$ and $\kappa_t^l = \sum_\tau \frac{\partial s_t^l}{\partial I_\tau^l}$, given the approximation error bound $\epsilon > 0$ s.t. $\left\|\mathbb{E}\left[\delta_t^{(s^l)}\kappa_t^l\right] - \mathbb{E}\left[\delta_t^{(s^l)}\right]\mathbb{E}\left[\kappa_t^l\right]\right\| \leq \epsilon(1 + \left\|\mathbb{E}\left[\delta_t^{(s^l)}\right]\right\|)$ for $\forall l$. Denote the stacked tensor $I^l = [I_1^l, ..., I_T^l]$ and $s^l = [s_1^l, ..., s_T^l]$. Assuming the backward procedure follows non-expansivity s.t. $\frac{\partial I^{l+1}}{\partial I^l} = W^{l+1^\top}\frac{\partial s^l}{\partial I^l}$ is 1-lipschitz continuous without loss of generality and the biases are bounded uniformly by $B$, i.e. $\left\|x\frac{\partial I^{l+1}}{\partial I^l} - \hat{x}\frac{\partial I^{l+1}}{\partial I^l}\right\| \leq \left\|x - \hat{x}\right\|$ for $\forall x, \hat{x}$. Define $\delta_{rate}^l = \left(\frac{\partial \mathcal{L}}{\partial c^l}\right)_{rate}$ as the error propagated through Eq. (7), and $\delta_t^{(I^l)} = \frac{\partial \mathcal{L}}{\partial I_t^l}$ as the error propagated through BPTT, with $\delta_{rate}^L = \mathbb{E}[\delta_t^{(I^L)}]$. We have the gradient difference bounded by $\left\|\delta_{rate}^{L-k} - \mathbb{E}[\delta_t^{(I^{L-k})}]\right\| = \mathcal{O}(k^2\epsilon)$.*

Theorem 2 elucidates the stability of rate-based backpropagation relative to BPTT, showing that the proposed method can provide a bound on the overall objective solution. The bounded error could further be interpreted as a form of randomness suitable for stochastic optimization. The similarity measurement of the descent directions between the two methods provides empirical evidence for the effectiveness of the proposed method (Figure 3d). Detailed proof is provided in Appendix A.

## 4.4 Implementation Details

For implementations of direct training, two distinct training modes are recognized: (multi-step) activation-based and (single-step) time-based [19], differing fundamentally in handling the timesteps loop. We implement our rate-based propagation in both modes: $\mathbf{rate}_M$ denotes the multi-step training mode where $T$ loops are embedded within layers, and $\mathbf{rate}_S$ refers to the single-step training mode with $T$ loops outside the layers. A detailed discussion of training modes is included in Appendix B.

Another aspect of our implementation concerns handling batch normalization (BN), especially given its critical role in BPTT, which adjusts mean and variance statistics during the forward pass. The application of BN varies depending on the training mode. In the multi-step mode, BN benefits from access to information across all timesteps and can normalize based on statistics aggregated over temporal dimensions. We employed tdBN [87] in $\mathbf{rate}_M$ since it has been widely adopted in direct training on various benchmarks. In contrast, the single-step mode limits BN to current timestep inputs, necessitating normalization across spatial dimensions only. In line with online schemes, SLTT [48] demonstrates the feasibility of implementing spatial BN iteratively across timesteps, an approach we adopt for $\mathbf{rate}_S$. Further details on the BN implementation are provided in Appendix B.

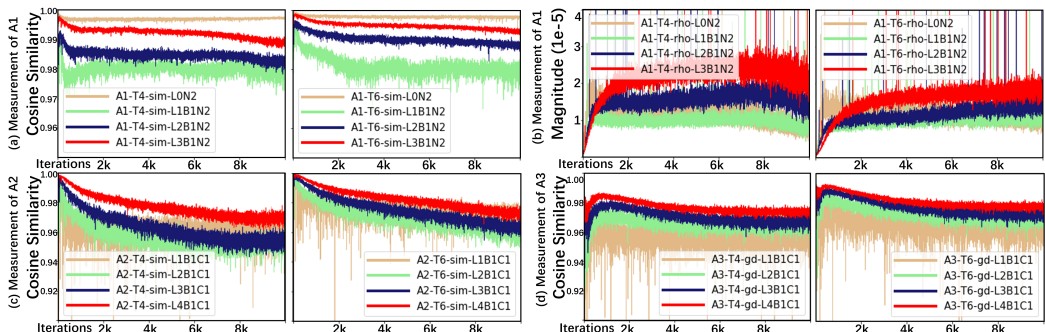

Figure 3: Empirical measurements conducted on the training procedure of BPTT. The experiments are carried out on the CIFAR-100 dataset using ResNet-18. Each subplot is labeled according to the naming convention "A{test#}-T{timesteps#}-{target}-L{layer#}B{block#}N{LIF#}/C{conv#}."

## 5    Experiments

In this section, we conduct experiments on CIFAR-10 [37], CIFAR-100 [37], ImageNet [11], and CIFAR10-DVS [41] to evaluate the proposed training method. We implement SNNs training on the Pytorch [53] and SpikingJelly [19] frameworks. We set $V_{th} = 1$, $\lambda = 0.2$, and employ the sigmoid-based surrogate function [19] for LIF neurons. Detailed setups are provided in Appendix C.

### 5.1    Empirical Validation

Empirical experiments are conducted to support the preconditions of theorems discussed in Section 4.3. These preconditions assert the independence of paired variables across the temporal dimension: $\mathbb{E}\left[\boldsymbol{\delta}_t^{(\boldsymbol{s}^l)} \boldsymbol{\kappa}_t^l\right] = \mathbb{E}\left[\boldsymbol{\delta}_t^{(\boldsymbol{s}^l)}\right] \mathbb{E}\left[\boldsymbol{\kappa}_t^l\right]$ (A1) and $\mathbb{E}\left[\boldsymbol{\delta}_t^{(\boldsymbol{I}^l)} \boldsymbol{s}_t^{l-1}\right] = \mathbb{E}\left[\boldsymbol{\delta}_t^{(\boldsymbol{I}^l)}\right] \mathbb{E}[\boldsymbol{s}_t^{l-1}]$ (A2). To explore these relationships, we conducted experiments training ResNet-18 on CIFAR-100 using BPTT. Cosine similarity measures were employed to compare the empirical expectation products, $cos\langle\mathbb{E}\left[\boldsymbol{\delta}_t^{(\boldsymbol{s}^l)} \boldsymbol{\kappa}_t^l\right], \mathbb{E}\left[\boldsymbol{\delta}_t^{(\boldsymbol{s}^l)}\right] \mathbb{E}\left[\boldsymbol{\kappa}_t^l\right]\rangle$ as shown in Figure 3a, where values approaching 1 indicate a high degree of alignment, suggesting that the variables' directions are similar. Additionally, the correlation coefficient, $\rho$ was measured to further assess the independence of these variables $\rho = \frac{\mathcal{COV}(\boldsymbol{\kappa}_t, \boldsymbol{\delta}_t^{(\boldsymbol{s}^l)})}{\sqrt{\mathrm{var}(\boldsymbol{\kappa}_t)\mathrm{var}(\boldsymbol{\delta}_t^{(\boldsymbol{s}^l)})}}$ where $\mathcal{COV}(\boldsymbol{\kappa}_t, \boldsymbol{\delta}_t^{(\boldsymbol{s}^l)}) = \mathbb{E}\left[\boldsymbol{\delta}_t^{(\boldsymbol{s}^l)} \boldsymbol{\kappa}_t^l\right] - \mathbb{E}\left[\boldsymbol{\delta}_t^{(\boldsymbol{s}^l)}\right] \mathbb{E}\left[\boldsymbol{\kappa}_t^l\right]$ It is clear that $\rho$ equals the cosine distance between the variables after centering by their means, $\rho = cos\langle\boldsymbol{\delta}_t^{(\boldsymbol{s}^l)} - \mathbb{E}[\boldsymbol{\delta}_t^{(\boldsymbol{s}^l)}], \boldsymbol{\kappa}_t^l - \mathbb{E}\left[\boldsymbol{\kappa}_t^l\right]\rangle$. Results, shown in Figure 3b, reveal that the correlation coefficients are constrained within a very small range, typically around the magnitude of $\sim 10^{-5}$, supporting the hypothesis of their relative independence. We also conducted cosine similarity measurements to validate the assumption $\mathbb{E}\left[\boldsymbol{\delta}_t^{(\boldsymbol{I}^l)} \boldsymbol{s}_t^{l-1}\right] = \mathbb{E}\left[\boldsymbol{\delta}_t^{(\boldsymbol{I}^l)}\right] \mathbb{E}[\boldsymbol{s}_t^{l-1}]$, as shown in Figure 3c. Additionally, we implement both BPTT and the proposed method simultaneously within the same training iteration, allowing direct observation of the gradient descent directions. The relation $(\nabla_{\boldsymbol{W}^l}\mathcal{L})_{\mathrm{rate}} = \frac{1}{T}(\nabla_{\boldsymbol{W}^l}\mathcal{L})$ (A3) was visualized in Figure 3d, which revealed that the convergence directions for rate-based backpropagation and BPTT are closely aligned. Remarkably, all tests consistently demonstrate that configurations with T=6 better adhere to the theoretical assumptions than T=4, suggesting that the proposed method can more closely mimic BPTT computations as the timestep increases. This observation also highlights the intrinsic link between our method and rate-coding, suggesting that a larger temporal window may facilitate more stable manifestations of rate-coding.

### 5.2    Comparison with the State-of-the-Art

We present comparison results in Table 1. In single-step mode, $\mathbf{rate}_S$ offers fair comparisons with online schemes, while $\mathbf{rate}_M$ in multi-step mode competes fairly with other methods employing one-step backpropagation. Unlike online methods such as OTTT[76], SLTT[48], and OS[89], which necessitate spatial backpropagation at every timestep, our proposed method conducts this process only once at the final timestep. Although methods of DSR [47] and SSF[68] delay decoupled backpropagation until the final timestep, allowing for parallel processing across all timesteps to enhance computational speed, they still require each timestep's backpropagation to be managed independently within the backward computation graph. In contrast, our method fully compresses the temporal dimension, achieving one-step time-independent spatial backpropagation. As shown in Table 1, our method yields comparable performance with BPTT counterparts on benchmarks, showcasing promising capabilities compared to other efficient training methods. While our theoretical

Table 1: Performance on CIFAR-10, CIFAR-100, ImageNet, and CIFAR10-DVS. Results are averaged over three runs of experiments, except for single crop evaluations on ImageNet. Models marked with (*) employ scaled weight standardization, adapting to normalizer-free architectures.

| | Training | Method | Model | Timesteps | Top-1 Acc (%) |
|---|---|---|---|---|---|
| CIFAR10 | QCFS [7] | ANN2SNN | ResNet-18 | 8 | 94.82 |
| | DSR [47] | one-step | PreAct-ResNet-18 | 20 | 95.10±0.15 |
| | SSF [68] | one-step | PreAct-ResNet-18 | 20 | 94.90 |
| | $BPTT_M$ | BPTT | ResNet-18 | 4 | 95.64 |
| | **$rate_M$ (ours)** | one-step | ResNet-18 | 4 | 95.61±0.02(95.64) |
| | OTTT [76] | online | VGG-11* | 6 | 93.52±0.06 |
| | SLTT [48] | online | ResNet-18 | 6 | 94.44±0.21 |
| | OS [89] | online | VGG-11 | 4 | 94.35 |
| | | | ResNet-19 | 4 | 95.20 |
| | $BPTT_S$ | BPTT | ResNet-18 | 4 | 95.53 |
| | | | VGG-11 | 4 | 95.61 |
| | **$rate_S$ (ours)** | one-step | ResNet-18 | 4 | 95.42±0.11(95.56) |
| | | | VGG-11 | 4 | 95.57±0.08(95.68) |
| CIFAR100 | DSR [47] | one-step | PreAct-ResNet-18 | 20 | 78.50±0.12 |
| | SSF [68] | one-step | PreAct-ResNet-18 | 20 | 75.48 |
| | $BPTT_M$ | BPTT | ResNet-18 | 4 | 77.93 |
| | **$rate_M$ (ours)** | one-step | ResNet-18 | 4 | 78.26±0.12(78.38) |
| | OTTT [76] | online | VGG-11* | 6 | 71.05±0.04 |
| | SLTT [48] | online | ResNet-18 | 6 | 74.38±0.30 |
| | OS [89] | online | VGG-11 | 4 | 76.48 |
| | | | ResNet-19 | 4 | 77.86 |
| | $BPTT_S$ | BPTT | ResNet-18 | 4 | 77.72 |
| | | | VGG-11 | 4 | 77.82 |
| | **$rate_S$ (ours)** | one-step | ResNet-18 | 4 | 77.73±0.28(77.93) |
| | | | VGG-11 | 4 | 77.87±0.35(78.13) |
| ImageNet | OTTT [76] | online | PreAct-ResNet-34* | 6 | 65.15 |
| | SLTT [48] | online | PreAct-ResNet-34* | 6 | 66.19 |
| | OS [89] | online | SEW-ResNet-34 | 4 | 64.14 |
| | | | PreAct-ResNet-34 | 4 | 67.54 |
| | SEW-ResNet [20] | BPTT | SEW-ResNet-34 | 4 | 67.04 |
| | **$rate_S$ (ours)** | one-step | SEW-ResNet-34 | 4 | 65.66 |
| | | | PreAct-ResNet-34 | 4 | 69.58 |
| | **$rate_M$ (ours)** | one-step | SEW-ResNet-34 | 4 | 65.84 |
| | | | PreAct-ResNet-34 | 4 | 70.01 |
| CIFAR10-DVS | DSR [47] | one-step | VGG-11 | 20 | 77.27±0.24 |
| | SSF [68] | | VGG-11 | 20 | 78.0 |
| | OTTT [76] | online | VGG-11* | 10 | 76.63±0.34 |
| | SLTT [48] | | VGG-11 | 10 | 77.17±0.23 |
| | $BPTT_S$ | BPTT | VGG-11 | 10 | 76.73 |
| | $BPTT_M$ | | VGG-11 | 10 | 76.86 |
| | **$rate_S$ (ours)** | one-step | VGG-11 | 10 | 76.48±0.23(76.71) |
| | **$rate_M$ (ours)** | | VGG-11 | 10 | 76.96±0.13(77.13) |

analysis and motivation primarily adhere to rate-coding approximations, the performance on static datasets aligns with expectations. The results on the dynamic dataset CIFAR10-DVS also achieve comparable levels, implying a significant presence of rate-based representation within CIFAR10-DVS. More results regarding the performance comparisons between the proposed method and BPTT across various architectures and settings have been detailed in Appendix D.

## 5.3 Impact of Time Expansion

we assess the impact of extending timesteps on both accuracy and training efficiency. Figure 4a validates that our method capably manages increased timesteps, thereby confirming the scalability of the proposed method for larger $T$ values. Figure 4b displays the computational and memory expenses incurred during the backward phase, which, as anticipated, do not escalate with increasing $T$.

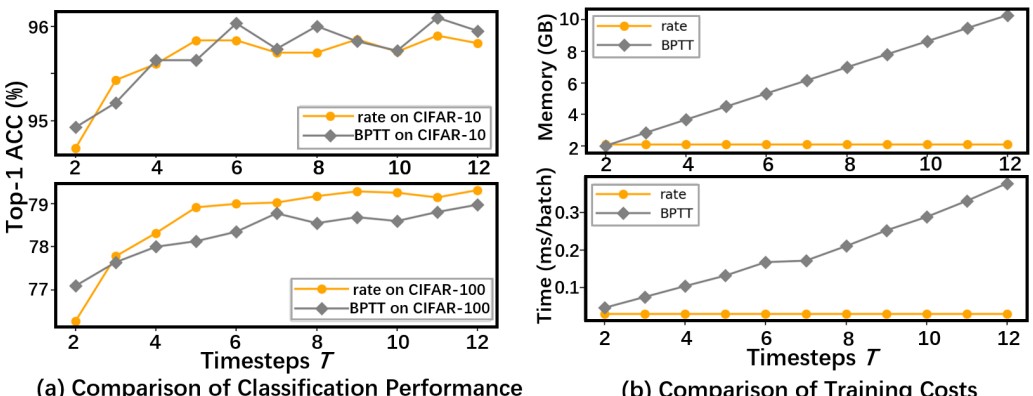

(a) Comparison of Classification Performance    (b) Comparison of Training Costs

Figure 4: Results of BPTT and $\mathbf{rate}_M$ across various timesteps.

Table 2: Performance w/o and w/ temporal shuffle for models trained by $\mathbf{rate}_M$

| Dataset | Model | Timesteps | Accuracy | Shuffled |
|---|---|---|---|---|
| CIFAR-10 | ResNet-18 | 2 | 94.77 | 94.63±0.04 |
| | | 4 | 95.51 | 95.50±0.04 |
| | | 6 | 95.97 | 95.95±0.09 |
| | VGG-11 | 2 | 95.13 | 95.10±0.05 |
| | | 4 | 95.37 | 95.37±0.03 |
| | | 6 | 95.77 | 95.79±0.05 |
| CIFAR-100 | ResNet-18 | 2 | 76.27 | 75.59±0.11 |
| | | 4 | 78.32 | 77.72±0.15 |
| | | 6 | 79.10 | 79.10±0.14 |
| | VGG-11 | 2 | 77.46 | 77.21±0.12 |
| | | 4 | 77.88 | 77.78±0.16 |
| | | 6 | 77.97 | 78.02±0.09 |
| ImageNet | SEW-ResNet-34 | 4 | 65.84 | 65.11±0.11 |
| | PreAct-ResNet-34 | 4 | 70.01 | 69.78±0.10 |
| CIFAR10-DVS | VGG-11 | 10 | 76.50 | 74.69±0.17 |

## 5.4 Analysis of Rate Statistics

Our method, derived from the principles of rate-based representation, necessitates examining the impact of rate coding on model behavior. Following an insightful approach from [6], we assess the robustness of our models by shuffling the temporal order of spike sequences while maintaining their rate consistency. This experiment, designed to disrupt temporal information without changing the firing rate, was applied to models trained using rate-based backpropagation. During inference on the test dataset, we introduced perturbations by randomly shuffling the temporal dimensions of input tensors across all neurons, as reported in Table 2. Notably, models mostly resisted these changes to some degree, which suggests that they follow the basic rules of rate coding, where the reordering of timesteps does not significantly impact overall accuracy. Furthermore, we tracked the average firing rates across each layer over time, presented in Figure 5. As layers increase, the average spike rates per layer are closely aligned with the temporal mean, validating the idea of rate-coding approximation. Those two experiments support the notion that rate-based backpropagation proficiently captures rate-based representations during training.

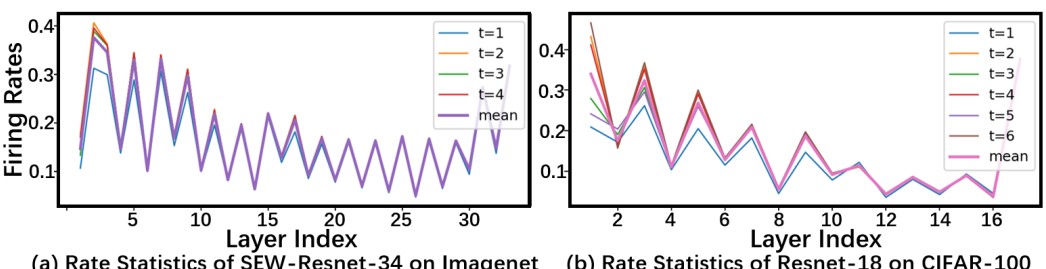

(a) Rate Statistics of SEW-Resnet-34 on Imagenet    (b) Rate Statistics of Resnet-18 on CIFAR-100

Figure 5: Firing rates statistics for models trained by $\mathbf{rate}_M$.

## 6 Conclusion

In this work, we propose rate-based backpropagation, utilizing rate-coding approximation to streamline the gradient computational graph, significantly reducing both memory usage and training time. Through theoretical analyses and empirical validation, we show the method's feasibility in approximating the optimization direction of BPTT. Experimental results across benchmarks reveal that our

method achieves comparable performance with BPTT and surpasses other state-of-the-art efficient training methods. We expect our work to pave the way for more scalable and resource-efficient training of SNNs.

## Acknowledgment

This work was supported by the National Natural Science Foundation of China (Grant No. 62304203), the Natural Science Foundation of Zhejiang Province, China (Grant No. LQ22F010011), and the ZJU-UIUC Center for Heterogeneously Integrated Brain-Inspired Computing.

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

# A Proof of Theorems

**Theorem 1.** *Given* $\delta_t^{(s^l)} = \frac{\partial \mathcal{L}}{\partial s_t^l}$ *that refers to gradients computed following the chain rule of BPTT in Eq. (2), and* $\kappa_t^l = \sum_\tau \frac{\partial s_t^l}{\partial I_\tau^l}$ *(where* $\mathbb{E}\left[\kappa_t^l\right] = \mathbb{E}\left[\varkappa_t^l\right]$ *in Eq.(6-7))* , *if* $\mathbb{E}\left[\delta_t^{(s^l)}\kappa_t^l\right] = \mathbb{E}\left[\delta_t^{(s^l)}\right]\mathbb{E}\left[\kappa_t^l\right]$ *holds for* $\forall l$*, we have* $\mathbb{E}\left[\delta_t^{(s^l)}\right] = \left(\frac{\partial \mathcal{L}}{\partial r^l}\right)_{rate}$ *. Furthermore, given* $\delta_t^{(I^l)} = \frac{\partial \mathcal{L}}{\partial I_t^l}$*, if* $\mathbb{E}\left[\delta_t^{(I^l)}s_t^{l-1}\right] = \mathbb{E}\left[\delta_t^{(I^l)}\right]\mathbb{E}[s_t^{l-1}]$ *for* $\forall l$*, we then obtain* $(\nabla_{W^l}\mathcal{L})_{rate} = \frac{1}{T}(\nabla_{W^l}\mathcal{L})$*. Here,* $\mathbb{E}\left[x_t\right] = \frac{1}{T}\sum_t x_t$ *refers the mean value of tensor* $x_t$ *over temporal dimension* $T$*.*

*Proof.* Given $\delta_t^{(s^l)} = \frac{\partial \mathcal{L}}{\partial s_t^l}$ and $\kappa_t^l = \sum_\tau \frac{\partial s_t^l}{\partial I_t^l}$, we establish the mean gradients through neural dynamics based on the chain rule:

$$\mathbb{E}\left[\frac{\partial \mathcal{L}}{\partial I_t^l}\right] = \mathbb{E}\left[\sum_\tau \frac{\partial \mathcal{L}}{\partial s_\tau^l}\frac{\partial s_\tau^l}{\partial I_t^l}\right] = \frac{1}{T}\sum_t\sum_\tau \frac{\partial \mathcal{L}}{\partial s_\tau^l}\frac{\partial s_\tau^l}{\partial I_t^l} = \mathbb{E}[\delta_t^{(s^l)}\kappa_t^l], \tag{10}$$

Considering the output layer $l = L$, the objective for BPTT can be expressed as $\mathcal{L} = \ell(\mathbb{E}[o_t], y) = \ell(\mathbb{E}[W^L s_t^L], y) = \ell(W^L\mathbb{E}[s_t^L], y) = \ell(W^L r^L, y)$. Under the rate-based objective $\mathcal{L} = \frac{1}{T}\ell(c^L, y) = \frac{1}{T}\ell(W^L r^{L-1}, y)$, it is clear that $\mathbb{E}[\delta_t^{(s^{L-1})}] = \left(\frac{\partial \mathcal{L}}{\partial r^{L-1}}\right)_{rate}$. Applying the precondition $\mathbb{E}[\delta_t^{(s^{L-1})}\kappa_t^L] = \mathbb{E}[\delta_t^{(s^{L-1})}]\mathbb{E}[\kappa_t^{L-1}]$, we obtain:

$$\begin{aligned}
\left(\frac{\partial \mathcal{L}}{\partial r^{L-1}}\frac{\partial r^{L-1}}{\partial c^{L-2}}\frac{\partial c^{L-2}}{\partial r^{L-2}}\right)_{rate} &= \left(\frac{\partial \mathcal{L}}{\partial r^{L-1}}\right)_{rate}\mathbb{E}[\kappa_t^{L-1}]W^{(L-1)^\top} \\
&= \mathbb{E}[\delta_t^{(s^{L-1})}]\mathbb{E}[\kappa_t^{L-1}]W^{(L-1)^\top} = \mathbb{E}[\delta_t^{(s^{L-1})}\kappa_t^{L-1}]W^{(L-1)^\top} \\
&= \mathbb{E}\left[\frac{\partial \mathcal{L}}{\partial I_t^{L-1}}\right]W^{(L-1)^\top} = \mathbb{E}\left[\frac{\partial \mathcal{L}}{\partial I_t^{L-1}}W^{(L-1)^\top}\right] \\
&= \mathbb{E}\left[\frac{\partial \mathcal{L}}{\partial s_t^{L-2}}\right] = \mathbb{E}[\delta_t^{(s^{L-2})}],
\end{aligned} \tag{11}$$

Continuing this induction process, we can derive that $\mathbb{E}[\delta_t^{(s^l)}] = \left(\frac{\partial \mathcal{L}}{\partial r^l}\right)_{rate}$ for all layers $l$. Further, given $\delta_t^{(I^l)} = \frac{\partial \mathcal{L}}{\partial I_t^l}$, the gradient for the weight matrix under BPTT: $\nabla_{W^l}\mathcal{L} = \sum_t\left(\frac{\partial \mathcal{L}}{\partial I_t^l}\frac{\partial I_t^l}{\partial W^l}\right) = \sum_t \delta_t^{(I^l)}s_t^{l-1}$. The gradients passing through the linear parts maintain the equivalence:

$$\begin{aligned}
\left(\frac{\partial \mathcal{L}}{\partial c^l}\right)_{rate} &= \left(\frac{\partial \mathcal{L}}{\partial r^{l+1}}\frac{\partial r^{l+1}}{\partial c^l}\right)_{rate} = \left(\frac{\partial \mathcal{L}}{\partial r^{l+1}}W^{l+1^\top}\right)_{rate} \\
&= \left(\frac{\partial \mathcal{L}}{\partial r^{l+1}}\right)_{rate}W^{l+1^\top} = \mathbb{E}[\delta_t^{(s^{l+1})}W^{l+1^\top}] = \mathbb{E}[\delta_t^{(I^l)}].
\end{aligned} \tag{12}$$

With the precondition that $\mathbb{E}[\delta_t^{(I^l)}s_t^{l-1}] = \mathbb{E}[\delta_t^{(I^l)}]\mathbb{E}[s_t^{l-1}]$ holds for $\forall l$, we obtain:

$$\begin{aligned}
(\nabla_{W^l}\mathcal{L})_{rate} &= \left(\frac{\partial \mathcal{L}}{\partial c^l}\frac{\partial c^l}{\partial W^l}\right)_{rate} = \left(\frac{\partial \mathcal{L}}{\partial c^l}\right)_{rate}r^{l-1} \\
&= \mathbb{E}[\delta_t^{(I^l)}]\mathbb{E}[s_t^{l-1}] = \mathbb{E}[\delta_t^{(I^l)}s_t^{l-1}] = \frac{1}{T}\nabla_{W^l}\mathcal{L}.
\end{aligned} \tag{13}$$

**Theorem 2.** *For gradients* $\delta_t^{(s^l)} = \frac{\partial \mathcal{L}}{\partial s_t^l}$ *and* $\kappa_t^l = \sum_\tau \frac{\partial s_t^l}{\partial I_\tau^l}$*, given the approximation error bound* $\epsilon > 0$ *s.t.* $\left\|\mathbb{E}\left[\delta_t^{(s^l)}\kappa_t^l\right] - \mathbb{E}\left[\delta_t^{(s^l)}\right]\mathbb{E}\left[\kappa_t^l\right]\right\| \leq \epsilon(1 + \left\|\mathbb{E}\left[\delta_t^{(s^l)}\right]\right\|)$ *for* $\forall l$*. Denote the stacked tensor* $I^l = [I_1^l, ..., I_T^l]$ *and* $s^l = [s_1^l, ..., s_T^l]$*. Assuming the backward procedure follows non-expansivity s.t.* $\frac{\partial I^{l+1}}{\partial I^l} = W^{l+1^\top}\frac{\partial s^l}{\partial I^l}$ *is 1-lipschitz continuous without loss of generality and the biases are bounded uniformly by B, i.e.* $\left\|x\frac{\partial I^{l+1}}{\partial I^l} - \hat{x}\frac{\partial I^{l+1}}{\partial I^l}\right\| \leq \left\|x - \hat{x}\right\|$ *for* $\forall x, \hat{x}$*. Define* $\delta_{rate}^l = \left(\frac{\partial \mathcal{L}}{\partial c^l}\right)_{rate}$ *as the*

*error propagated through Eq. (7), and $\boldsymbol{\delta}_t^{(\boldsymbol{I}^l)} = \frac{\partial \mathcal{L}}{\partial \boldsymbol{I}_t^l}$ as the error propagated through BPTT, with $\boldsymbol{\delta}_{rate}^L = \mathbb{E}[\boldsymbol{\delta}_t^{(\boldsymbol{I}^L)}]$. We have the gradient difference bounded by $\left\| \boldsymbol{\delta}_{rate}^{L-k} - \mathbb{E}[\boldsymbol{\delta}_t^{(\boldsymbol{I}^{L-k})}] \right\| = \mathcal{O}(k^2 \epsilon)$.*

*Proof.* Given that the error backpropagation $\frac{\partial \boldsymbol{I}^{l+1}}{\partial \boldsymbol{I}^l}$ follows a 1-Lipschitz condition with biases bounded by $B$ for all $l$, we can derive $\left\| \mathbb{E}[\boldsymbol{\delta}_t^{(\boldsymbol{I}^l)}] \right\| = \left\| \mathbb{E}[\boldsymbol{\delta}_t^{(\boldsymbol{I}^{l+1})} \frac{\partial \boldsymbol{I}^{l+1}}{\partial \boldsymbol{I}^l}] \right\| \leq \left\| \mathbb{E}[\boldsymbol{\delta}_t^{(\boldsymbol{I}^{l+1})}] \right\| + B$ by non-expansivity. Then, by induction, we obtain the gradient bound between the intermediate layers and the final layer:

$$\left\| (\mathbb{E}[\boldsymbol{\delta}_t^{(\boldsymbol{I}^l)}]) \right\| \leq (L-l)B + \left\| \mathbb{E}[\boldsymbol{\delta}_t^{(\boldsymbol{I}^L)}] \right\|.$$

Since $\frac{\partial \boldsymbol{I}^{l+1}}{\partial \boldsymbol{I}_t^l} = \boldsymbol{W}^{l+1\top} \boldsymbol{\kappa}_t^l$ is also 1-Lipschitz continuous without loss of generality, given the approximated approximated error $\epsilon > 0$ s.t.

$$
\begin{aligned}
\left\| \mathbb{E}[\boldsymbol{\delta}_t^{(\boldsymbol{I}^{l+1})}] \boldsymbol{W}^{l+1\top} \mathbb{E}[\boldsymbol{\kappa}_t^l] - \mathbb{E}[\boldsymbol{\delta}_t^{(\boldsymbol{I}^{l+1})} \boldsymbol{W}^{l+1\top} \boldsymbol{\kappa}_t^l] \right\| &= \left\| \mathbb{E}[\boldsymbol{\delta}_t^{(\boldsymbol{s}^l)}] \mathbb{E}[\boldsymbol{\kappa}_t^l] - \mathbb{E}[\boldsymbol{\delta}_t^{(\boldsymbol{s}^l)} \boldsymbol{\kappa}_t^l] \right\| \\
&\leq \epsilon(1 + \left\| \mathbb{E}[\boldsymbol{\delta}_t^{(\boldsymbol{s}^l)} \boldsymbol{\kappa}_t^l] \right\|) = \epsilon(1 + \left\| \mathbb{E}[\boldsymbol{\delta}_t^{(\boldsymbol{I}^{l+1})} \boldsymbol{W}^{l+1\top} \boldsymbol{\kappa}_t^l] \right\|)
\end{aligned}
\tag{14}
$$

we have

$$
\begin{aligned}
\left\| \boldsymbol{\delta}_{\text{rate}}^l - \mathbb{E}[\boldsymbol{\delta}_t^{(\boldsymbol{I}^l)}] \right\| &= \left\| \boldsymbol{\delta}_{\text{rate}}^{l+1} \boldsymbol{W}^{l+1\top} \mathbb{E}[\boldsymbol{\kappa}_t^l] - \mathbb{E}[\boldsymbol{\delta}_t^{(\boldsymbol{I}^{l+1})}] \boldsymbol{W}^{l+1\top} \boldsymbol{\kappa}_t^l \right\| \\
&= \left\| \left( \boldsymbol{\delta}_{\text{rate}}^{l+1} \boldsymbol{W}^{l+1\top} \mathbb{E}[\boldsymbol{\kappa}_t^l] - \mathbb{E}[\boldsymbol{\delta}_t^{(\boldsymbol{I}^{l+1})}] \boldsymbol{W}^{l+1\top} \mathbb{E}[\boldsymbol{\kappa}_t^l] \right) \right. \\
&\quad \left. + \left( \mathbb{E}[\boldsymbol{\delta}_t^{(\boldsymbol{I}^{l+1})}] \boldsymbol{W}^{l+1\top} \mathbb{E}[\boldsymbol{\kappa}_t^l] - \mathbb{E}[\boldsymbol{\delta}_t^{(\boldsymbol{I}^{l+1})} \boldsymbol{W}^{l+1\top} \boldsymbol{\kappa}_t^l] \right) \right\| \\
&\leq \left\| \boldsymbol{\delta}_{\text{rate}}^{l+1} - \mathbb{E}[\boldsymbol{\delta}_t^{(\boldsymbol{I}^{l+1})}] \right\| + \epsilon(1 + \left\| \mathbb{E}[\boldsymbol{\delta}_t^{(\boldsymbol{I}^{l+1})} \boldsymbol{W}^{l+1\top} \boldsymbol{\kappa}_t^l] \right\|) \\
&\leq \left\| \boldsymbol{\delta}_{\text{rate}}^{l+1} - \mathbb{E}[\boldsymbol{\delta}_t^{(\boldsymbol{I}^{l+1})}] \right\| + \epsilon(1 + (L-l)B + \left\| \mathbb{E}[\boldsymbol{\delta}_t^{(\boldsymbol{I}^L)}] \right\|)
\end{aligned}
\tag{15}
$$

By induction, we obtain

$$\left\| \boldsymbol{\delta}_{\text{rate}}^l - \mathbb{E}[\boldsymbol{\delta}_t^{(\boldsymbol{I}^l)}] \right\| \leq \epsilon \left( (L-l) + \frac{(L-l+1)(L-l)}{2}B + (L-l) \left\| \mathbb{E}[\boldsymbol{\delta}_t^{(\boldsymbol{I}^L)}] \right\| \right) = \mathcal{O}\left( (L-l)^2 \epsilon \right) \tag{16}$$

## B   Implementation Details

### B.1   Pseudocode of the Rate-based Backpropagation

The pseudocode for rate-based backpropagation, illustrating the implementations for both $\textbf{rate}_M$ and $\textbf{rate}_S$, is provided in Algorithm 1.

### B.2   About Training Modes in Rate-based Backpropagation

In direct training, two distinct implementation modes are recognized, activation-based and time-based [19], differing fundamentally in their handling of the simulation timestep $T$. The activation-based, also known as multi-step mode, processes the $T$ loop separately within each layer, transmitting inter-layer tensors within dimensions $[T, B, S]$, where $B$ and $S$ refer to batch and spatial dimensions, respectively. The configuration enables the multi-step mode to enhance computational efficiency by reformatting the tensor dimensions as $[T \times B, S]$ to optimize parallelism in linear parts. However, the coupled processing with temporal calculations embedded within the layers increases memory retention on GPUs, potentially obscuring the benefits of memory cost optimization in both online training and our proposed methods. In contrast, the time-based mode externalizes the $T$ loop, facilitating single-step forward computations at each timestep. This single-step mode aligns well with the dynamic modeling of temporal dimensions and facilitates memory optimization strategies more effectively. However, its restriction on parallel computation in linear components compared to multi-step mode necessitates increased forward time on GPUs, albeit with enhanced support for memory optimization. Our proposed method has been adapted to operate effectively within both frameworks to ensure comprehensiveness, as shown in Algorithm 1.

**Algorithm 1:** Single Training Iteration of the Rate-based Backpropagation

**Input:** Timesteps $T$; Network depth $L$; Trainable parameters $\{\boldsymbol{W}^l\}_{l \leq L}$; Training Mini-batch $\{(\boldsymbol{x}_t^0, \boldsymbol{y})\}$; Training Mode ***rate***$_S$ or ***rate***$_M$.

**Output:** Updated parameters $\{\boldsymbol{W}^l\}_{l \leq L}$

1   Initialize input spikes $\boldsymbol{s}_t^0 = \boldsymbol{x}_t^0$ for all $t \in [1, T]$.

2   **if** ***rate***$_M$ **then**

3      **for** $l = 1$ *to* $L$ **do**

4         Compute input currents through linear operators $\boldsymbol{I}_t^l = \boldsymbol{W}^l \boldsymbol{s}_t^{l-1}$ for all $t \in [1, T]$;

5         Initialize $\boldsymbol{\rho}_0^l = 0$, $\boldsymbol{g}_0^l = 0$, $\boldsymbol{e}_0^l = 0$.

6         **for** $t = 1$ *to* $T$ **do**

7            Compute output spikes $\boldsymbol{s}_t^l$ from $\boldsymbol{I}_t^l$ following neural dynamics in Eq. (1);

8            Compute the eligibility trace $\boldsymbol{\rho}_t^l = 1 + \boldsymbol{\rho}_{t-1}^l \left( \frac{\partial \boldsymbol{u}_t^l}{\partial \boldsymbol{u}_{t-1}^l} + \frac{\partial \boldsymbol{u}_t^l}{\partial \boldsymbol{s}_{t-1}^l} \frac{\partial \boldsymbol{s}_{t-1}^l}{\partial \boldsymbol{u}_{t-1}^l} \right)$ in Eq. (8);

9            Accumulate $\boldsymbol{e}_t^l = \frac{1}{t}((t-1)\boldsymbol{e}_{t-1}^l + \boldsymbol{s}_t^l)$;

10            Accumulate $\boldsymbol{g}_t^l = \frac{1}{t}((t-1)\boldsymbol{g}_{t-1}^l + \frac{\partial \boldsymbol{s}_t^l}{\partial \boldsymbol{u}_t^l}\boldsymbol{\rho}_t)$.

11         **end**

12         Save $\boldsymbol{e}_T^l$, $\boldsymbol{g}_T^l$ and $\boldsymbol{W}^l$ for backwards, and free intermediate variables.

13      **end**

14   **else**

15      Initialize $\boldsymbol{\rho}_0^l = 0$, $\boldsymbol{g}_0^l = 0$, $\boldsymbol{e}_0^l = 0$ for all $l \in [1, L]$.

16      **for** $t = 1$ *to* $T$ **do**

17         **for** $l = 1$ *to* $L$ **do**

18            Compute input currents through linear operators $\boldsymbol{I}_t^l = \boldsymbol{W}^l \boldsymbol{s}_t^{l-1}$;

19            Initialize $\boldsymbol{\rho}_0^l = 0$, $\boldsymbol{g}_0^l = 0$, $\boldsymbol{e}_0^l = 0$;

20            Compute output spikes $\boldsymbol{s}_t^l$ from $\boldsymbol{I}_t^l$ following neural dynamics in Eq. (1);

21            Compute the eligibility trace $\boldsymbol{\rho}_t^l = 1 + \boldsymbol{\rho}_{t-1}^l \left( \frac{\partial \boldsymbol{u}_t^l}{\partial \boldsymbol{u}_{t-1}^l} + \frac{\partial \boldsymbol{u}_t^l}{\partial \boldsymbol{s}_{t-1}^l} \frac{\partial \boldsymbol{s}_{t-1}^l}{\partial \boldsymbol{u}_{t-1}^l} \right)$ in Eq. (8);

22            Accumulate $\boldsymbol{e}_t^l = \frac{1}{t}((t-1)\boldsymbol{e}_{t-1}^l + \boldsymbol{s}_t^l)$;

23            Accumulate $\boldsymbol{g}_t^l = \frac{1}{t}((t-1)\boldsymbol{g}_{t-1}^l + \frac{\partial \boldsymbol{s}_t^l}{\partial \boldsymbol{u}_t^l}\boldsymbol{\rho}_t)$;

24            Save $\boldsymbol{u}_t^l$, $\boldsymbol{s}_t^l$ for neuron states;

25            Save $\boldsymbol{g}_t^l$, $\boldsymbol{e}_t^l$, $\boldsymbol{\rho}_t^l$ as eligibility traces.

26         **end**

27      **end**

28   **end**

29   Compute the outputs gradient $\frac{\partial \mathcal{L}}{\partial \boldsymbol{c}^L}$ from the objective function.

30   **for** $l = L - 1$ *to* $1$ **do**

31      Compute error backpropagated through the linear part $\frac{\partial \mathcal{L}}{\partial \boldsymbol{r}^l} = \frac{\partial \mathcal{L}}{\partial \boldsymbol{c}^{l+1}}\boldsymbol{W}^{l+1\top}$;

32      Compute error backpropagated through the neuron part $\frac{\partial \mathcal{L}}{\partial \boldsymbol{c}^l} = \frac{\partial \mathcal{L}}{\partial \boldsymbol{r}^l}\boldsymbol{g}_T^l$;

33      Compute the weight gradients $\nabla_{\boldsymbol{W}^l}\mathcal{L} = \frac{\partial \mathcal{L}}{\partial \boldsymbol{c}^l}(\boldsymbol{e}_T^{l-1})^\top$;

34      Update parameters $\{\boldsymbol{W}^l\}_{l \leq L}$ based on the gradient-based optimizer.

35   **end**

## B.3   Implementation of Batch Normalization in Rate-based Backpropagation

In the forward pass, batch normalization (BN) precedes neuron activation, scaling inputs $\boldsymbol{I}_t^l$ and introduces a bias in the average inputs $\boldsymbol{c} = \mathbb{E}[\boldsymbol{I}_t]$. We denote $\tilde{\boldsymbol{c}}$ to represent the biased average inputs as $\tilde{\boldsymbol{c}} = \mathbb{E}[\tilde{\boldsymbol{I}}_t] = \mathbb{E}[\text{BN}(\boldsymbol{I}_t)]$ instead of $\boldsymbol{c}$. Note that BN acts as a linear operation during inference, where $\tilde{\boldsymbol{c}} = \mathbb{E}[\tilde{\boldsymbol{I}}_t] = \mathbb{E}[\text{BN}(\boldsymbol{I}_t)] = \text{BN}(\mathbb{E}[\boldsymbol{I}_t]) = \text{BN}(\boldsymbol{c})$. Implementing rate-based propagation requires considering how gradients pass through the BN layers and affect their intrinsic parameters during training. Initially, we explore the spatial BN [48] design for the single-step mode, which

computes mean and variance statistics independently at each time step $t$:

$$\tilde{I}_t = \mathrm{BN}(I_t) = \gamma\left(\frac{I_t - \mu_t}{\sqrt{\sigma_t^2 + \epsilon}}\right) + \beta, \text{ where } \mu_t = \frac{1}{B}\sum_b I_t^{(b)} \text{ and } \sigma_t^2 = \frac{1}{B}\sum_b (I_t^{(b)} - \mu_t)^2. \quad (17)$$

Defining $\chi_t^{(I)} = \frac{\partial \tilde{I}_t}{\partial I_t}, \chi_t^{(\gamma)} = \frac{\partial \tilde{I}_t}{\partial \gamma}, \chi_t^{(\beta)} = \frac{\partial \tilde{I}_t}{\partial \beta}$, the following expressions are obtained:

$$\chi_t^{(I)} = \gamma\frac{1}{\sqrt{\sigma_t^2 + \epsilon}} + \frac{\partial \tilde{I}_t^l}{\partial \sigma_t^2}\frac{\partial \sigma_t^2}{\partial I_t^l} + \frac{\partial \tilde{I}_t^l}{\partial \mu_t}\frac{\partial \mu_t}{\partial I_t^l}, \quad \chi_t^{(\gamma)} = \frac{I_t^l - \mu_t}{\sqrt{\sigma_t^2 + \epsilon}}, \quad \chi_t^{(\beta)} = Id. \quad (18)$$

For the backward derivation of BN in a rate-based setting based on mean estimations through time, we implement $\frac{\partial \mathcal{L}}{\partial c} = \frac{\partial \mathcal{L}}{\partial \tilde{c}}\mathbb{E}[\chi_t^{(I)}], \frac{\partial \mathcal{L}}{\partial \gamma} = \frac{\partial \mathcal{L}}{\partial \tilde{c}}\mathbb{E}[\chi_t^{(\gamma)}], \frac{\partial L}{\partial \beta} = \frac{\partial \mathcal{L}}{\partial \tilde{c}}\mathbb{E}[\chi_t^{(\beta)}] = \frac{\partial \mathcal{L}}{\partial \tilde{c}}$. Since gradient computation at each timestep is independent, the dynamic estimations of $\mathbb{E}[\chi_t^{(I)}]$ and $\mathbb{E}[\chi_t^{(\gamma)}]$ are performed in the same manner of $\{e_t^l\}_{t \leq T}$ and $\{g_t^l\}_{t \leq T}$.

In the multi-step mode, tdBN [87] accounts for mean and variance statistics over the entire time horizon:

$$\tilde{I}_t = \mathrm{BN}(I_t) = \gamma\left(\frac{I_t - \mu}{\sqrt{\sigma^2 + \epsilon}}\right) + \beta, \text{where } \mu = \frac{1}{BT}\sum_t\sum_b I_t^{(b)}, \sigma^2 = \frac{1}{BT}\sum_t\sum_b (I_t^{(b)} - \mu)^2. \quad (19)$$

The rate-based representation integrates the input across the time dimension, with the mean $\mu_c = \sum_b c^{(b)}$, and variance $\sigma_c^2 = \frac{1}{B}\sum_b (c^{(b)} - \mu_c)^2$. Since $c$ is the temporal mean of inputs, it is clear that $\mu_c = \mu$ and $\sigma_c^2 \leq \sigma^2$. Note that $\frac{\partial \sigma^2}{\partial I_t} = \frac{1}{BT}\sum_t\sum_b(I_t^{(b)} - \mu) = \frac{1}{B}\sum_b(c^{(b)} - \mu_c) = \frac{\partial \sigma_c^2}{\partial c}$. Assuming $\frac{\partial I_t}{\partial c} = Id$, we derive $\frac{\partial \mu}{\partial I_t} = \frac{\partial \mu_c}{\partial c}$. For the forward approximation specifically tailored for tdBN in rate-based backpropagation, we define:

$$\tilde{c} = \hat{\mathrm{BN}}(c) = \gamma\left(\frac{c - \mu}{\sqrt{\hat{\sigma}_c^2 + \epsilon}}\right) + \beta, \quad (20)$$

where $\gamma$ and $\beta$ refer to the same intrinsic parameters shared with $\mathrm{BN}(I_t)$, and $\hat{\sigma}_c^2$ is defined distinctly in forward and backward passes: $\hat{\sigma}_c^2 = \sigma^2$ in forward and $\frac{\partial \hat{\sigma}_c^2}{\partial \sigma_c^2} = Id$ in backward. The implementation utilizes gradient replacement with the detach operation in PyTorch: $\hat{\sigma}_c^2 = detach(\sigma^2 - \sigma_c^2) + \sigma_c^2$. Thus, in the forward phase, $\tilde{c} = \hat{\mathrm{BN}}(c) = \mathbb{E}[\mathrm{BN}(I_t)]$, and in the backward phase, $\frac{\partial \tilde{c}}{\partial c} = \mathbb{E}[\frac{\partial \tilde{c}}{\partial I_t}]$, aligning perfectly with the foundational principles of rate-based backpropagation.

## C  Experimental Settings

### C.1  Datasets

**CIFAR-10 and CIFAR-100.** The CIFAR-10 and CIFAR-100 [37] datasets contain 32x32 color images across different classes, licensed under MIT. CIFAR-10 includes 60,000 images across 10 classes, with 50,000 for training and 10,000 for testing, whereas CIFAR-100 is spread over 100 classes. Both datasets have been normalized for zero mean and unit variance. Image data augmentation is applied using AutoAugment [9] and Cutout [14] strategies, similar to the implementations in recent studies [42, 7, 24, 69, 13]. The pixel values are directly fed into the input layer at each timestep as direct encoding [55].

**ImageNet.** The ImageNet-1K dataset [11] comprises 1,281,167 training images and 50,000 validation images distributed across 1,000 classes, licensed for non-commercial use. ImageNet-1K images are normalized for zero mean and unit variance. Training images undergo random resized cropping to 224x224 pixels and horizontal flipping, while validation images are resized to 256x256 and then center-cropped to 224x224. The images are transformed into time sequences through direct encoding [55], following the approach used for CIFAR datasets.

**CIFAR10-DVS.** The CIFAR10-DVS dataset [41] is a neuromorphic version of CIFAR-10, which includes 10,000 event-based images captured by the DVS camera with pixel dimensions expanded to

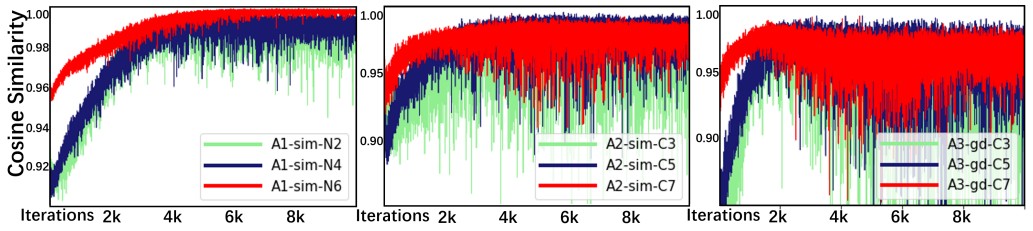

Figure 6: Empirical measurements conducted on the CIFAR10-DVS dataset.

128×128, licensed under CC BY 4.0. We split the whole dataset into 9000 training images and 1000 testing images. Data preprocessing involves integrating events into frames [21, 19] and reducing the spatial resolution to 48x48 through interpolation. Additional data augmentation includes random horizontal flips and random rolls within a 5-pixel range, mirroring previous methods [76, 48].

## C.2 Training Setup

**Network Architectures**. For the CIFAR-10, CIFAR-100, and CIFAR10-DVS datasets, our method is tested on standard network architectures, including ResNet-18, ResNet-19, and VGG-11 [63, 32, 87, 76, 19, 69]. On the ImageNet dataset, we adapt two variations on ResNet architecture [32], SEW-ResNet-34 [87] specially proposed for SNNs, and ResNet-34 with pre-activation residual blocks [33], aligning with previous works [76, 48, 89]. While OTTT [76] and SLTT [48] frameworks utilize normalization-free techniques under the ResNet-34 framework [5], Zhu et al. [89] substitute these with their custom-designed batch normalization. We directly employ tdBN [87] instead of normalization-free methods in our experiments.

**Training Details.** This work utilizes the widely adopted sigmoid-based surrogate gradient [19] to approximate the Heaviside step function using $h(x, \alpha) = \frac{1}{1+e^{\alpha x}}$ and sets $\alpha = 4$ to ensure the maximum derivative of the surrogate function is 1 for preventing gradient explosion. All implementations are based on the PyTorch [53] and SpikingJelly [19] frameworks. The experiments on CIFAR-10, CIFAR-100, and CIFAR10-DVS datasets run on one NVIDIA GeForce RTX 3090 GPU. For ImageNet, distributed data parallel processing is utilized across eight NVIDIA GeForce RTX 4090 GPUs. We use the SGD optimizer [58] with a momentum of 0.9 for all tasks, integrating a cosine annealing strategy [45] for the learning rate schedule. Other hyperparameters are listed in Table 3.

Table 3: Training hyperparameters.

|  | CIFAR-10 | CIFAR-100 | ImageNet | CIFAR10-DVS |
|---|---|---|---|---|
| Epoch | 300 | 300 | 100 | 300 |
| Learning rate | 0.1 | 0.1 | 0.2 | 0.1 |
| Batch size | 128 | 128 | 512 | 128 |
| Weight decay | 5e-4 | 5e-4 | 2e-5 | 5e-4 |

# D   More Results

## D.1   Empirical Validation on CIFAR10-DVS

As shown in Figure 6, we extend conduct empirical experiments on CIFAR10-DVS as avalidation in the case of dynamic datasets. The observations confirm that, even in data with a degree of temporal information, the empirical validation of the assumptions remains consistent with expectations. This alignment emphasizes that the approximate relationship between rate-based backpropagation and BPTT remains substantially consistent. As a result, this stability ensures that our approach continues to effectively extract rate-based representations from neuromorphic datasets with a degree of temporal dynamics, thereby maintaining robust performance across diverse data scenarios.

## D.2   Extended Performance Comparisons with BPTT

We conduct additional experiments to illustrate the comparative performance of rate-based backpropagation versus BPTT, as presented in Table 4 for CIFAR-10 and Table 5 for CIFAR-100. These experiments span various configurations, including different network architectures—ResNet-18,

Table 4: Performance comparison of rate-based backpropagation and BPTT on CIFAR-10.

| Training | Model | Timesteps | Top-1 Acc (%) |
|---|---|---|---|
| $\text{BPTT}_S$ | ResNet-18 | 2 | 95.02 |
| | | 4 | 95.53 |
| | | 6 | 95.68 |
| | ResNet-19 | 2 | 96.12 |
| | | 4 | 96.38 |
| | | 6 | 96.57 |
| | VGG-11 | 2 | 95.27 |
| | | 4 | 95.61 |
| | | 6 | 95.63 |
| $\textbf{rate}_S$ | ResNet-18 | 2 | 94.82±0.07(94.89) |
| | | 4 | 95.42±0.11(95.56) |
| | | 6 | 95.73±0.03(95.78) |
| | ResNet-19 | 2 | 96.11±0.05(96.18) |
| | | 4 | 96.32±0.04(96.38) |
| | | 6 | 96.38±0.06(96.45) |
| | VGG-11 | 2 | 95.44±0.02(95.46) |
| | | 4 | 95.57±0.08(95.68) |
| | | 6 | 95.64±0.12(95.76) |
| $\text{BPTT}_M$ | ResNet-18 | 2 | 94.93 |
| | | 4 | 95.64 |
| | | 6 | 96.03 |
| | ResNet-19 | 2 | 96.16 |
| | | 4 | 96.49 |
| | | 6 | 96.70 |
| | VGG-11 | 2 | 95.31 |
| | | 4 | 95.67 |
| | | 6 | 95.64 |
| $\textbf{rate}_M$ | ResNet-18 | 2 | 94.75±0.05(94.82) |
| | | 4 | 95.61±0.02(95.64) |
| | | 6 | 95.90±0.07(96.01) |
| | ResNet-19 | 2 | 96.23±0.10(96.33) |
| | | 4 | 96.26±0.03(96.29) |
| | | 6 | 96.38±0.02(96.40) |
| | VGG-11 | 2 | 95.17±0.12(95.35) |
| | | 4 | 95.30±0.06(95.37) |
| | | 6 | 95.23±0.06(95.32) |

Table 5: Performance comparison of rate-based backpropagation and BPTT on CIFAR-100.

| Training | Model | Timesteps | Top-1 Acc (%) |
|---|---|---|---|
| $\mathbf{BPTT}_S$ | ResNet-18 | 2 | 76.24 |
| | | 4 | 77.72 |
| | | 6 | 78.65 |
| | ResNet-19 | 2 | 79.33 |
| | | 4 | 80.12 |
| | | 6 | 80.77 |
| | VGG-11 | 2 | 77.37 |
| | | 4 | 77.82 |
| | | 6 | 78.13 |
| $\mathbf{rate}_S$ | ResNet-18 | 2 | 75.89±0.11(75.97) |
| | | 4 | 77.73±0.28(77.93) |
| | | 6 | 78.86±0.08(78.94) |
| | ResNet-19 | 2 | 79.71±0.02(79.74) |
| | | 4 | 80.41±0.14(80.54) |
| | | 6 | 80.75±0.05(80.79) |
| | VGG-11 | 2 | 77.34±0.04(77.37) |
| | | 4 | 77.87±0.35(78.13) |
| | | 6 | 78.23±0.03(78.27) |
| $\mathbf{BPTT}_M$ | ResNet-18 | 2 | 77.09 |
| | | 4 | 77.93 |
| | | 6 | 78.35 |
| | ResNet-19 | 2 | 80.01 |
| | | 4 | 81.07 |
| | | 6 | 81.12 |
| | VGG-11 | 2 | 77.42 |
| | | 4 | 77.96 |
| | | 6 | 78.25 |
| $\mathbf{rate}_M$ | ResNet-18 | 2 | 75.97±0.20(76.27) |
| | | 4 | 78.26±0.12(78.38) |
| | | 6 | 79.02±0.11(79.16) |
| | ResNet-19 | 2 | 79.87±0.03(79.90) |
| | | 4 | 80.71±0.12(80.84) |
| | | 6 | 80.83±0.07(80.94) |
| | VGG-11 | 2 | 77.40±0.05(77.46) |
| | | 4 | 77.86±0.03(77.89) |
| | | 6 | 77.99±0.11(78.11) |

Table 6: Comparison results of performance and training costs across various timesteps. All units for time measurements are in seconds per batch. Experiments were conducted on NVIDIA GeForce RTX 4090, with training settings consistent with other experiments.

| Datasets | Network | Method | | Timesteps | | | | |
|---|---|---|---|---|---|---|---|---|
| | | | | T=1 | T=2 | T=4 | T=8 | T=16 |
| CIFAR100 | ResNet-18 | $\mathbf{rate}_M$ | Time of Eligibility Track | 0.003 | 0.004 | 0.007 | 0.015 | 0.027 |
| | | | Time of Backward | 0.034 | 0.035 | 0.036 | 0.034 | 0.036 |
| | | | Time of both | 0.037 | 0.039 | 0.043 | 0.049 | 0.063 |
| | | | Memory Allocated | 1.8492 | 1.8488 | 1.8473 | 1.8496 | 1.8483 |
| | | | Top-1 Acc [%] | 74.60 | 76.04 | 78.24 | 79.24 | 79.37 |
| | | $\mathrm{BPTT}_M$ | Time of Backward | 0.023 | 0.044 | 0.098 | 0.199 | 0.564 |
| | | | Memory Allocated | 1.4272 | 2.4454 | 4.4804 | 8.0460 | 15.685 |
| | | | Top-1 Acc [%] | 74.38 | 76.65 | 78.49 | 78.35 | |
| | ResNet-19 | $\mathbf{rate}_M$ | Time of Eligibility Track | 0.006 | 0.012 | 0.020 | 0.041 | |
| | | | Time of Backward | 0.083 | 0.083 | 0.082 | 0.083 | |
| | | | Time of both | 0.089 | 0.095 | 0.102 | 0.124 | |
| | | | Memory Allocated [GB] | 4.4787 | 4.4798 | 4.4788 | 4.4784 | |
| | | | Top-1 Acc [%] | 78.3 | 80.00 | 80.65 | 81.31 | |
| | | $\mathrm{BPTT}_M$ | Time of Backward | 0.046 | 0.111 | 0.285 | 0.552 | |
| | | | Memory Allocated [GB] | 3.2556 | 5.6636 | 10.8978 | 20.3862 | |
| | | | Top-1 Acc [%] | 78.39 | 80.06 | 81.11 | 81.13 | |
| | VGG11 | $\mathbf{rate}_M$ | Time of Eligibility Track | 0.003 | 0.003 | 0.006 | 0.011 | 0.020 |
| | | | Time of Backward | 0.017 | 0.017 | 0.017 | 0.017 | 0.018 |
| | | | Time of both | 0.020 | 0.020 | 0.023 | 0.028 | 0.038 |
| | | | Memory Allocated [GB] | 1.3624 | 1.3607 | 1.3619 | 1.3613 | 1.3601 |
| | | | Top-1 Acc [%] | 76.13 | 77.59 | 77.75 | 78.34 | 78.65 |
| | | $\mathrm{BPTT}_M$ | Time of Backward | 0.010 | 0.021 | 0.054 | 0.135 | 0.384 |
| | | | Memory Allocated [GB] | 0.9911 | 1.6784 | 3.7363 | 6.6141 | 12.3768 |
| | | | Top-1 Acc [%] | 76.34 | 77.20 | 77.98 | 78.26 | 78.37 |
| ImageNet | SEW-ResNet-34 | $\mathbf{rate}_M$ | Time of Eligibility Track | 0.012 | 0.014 | 0.023 | | |
| | | | Time of Backward | 0.074 | 0.074 | 0.074 | | |
| | | | Time of both | 0.086 | 0.088 | 0.097 | | |
| | | | Memory Allocated [GB] | 5.7887 | 5.7898 | 5.7883 | | |
| | | $\mathrm{BPTT}_M$ | Time of Backward | 0.046 | 0.095 | 0.233 | | |
| | | | Memory Allocated [GB] | 3.9858 | 6.8654 | 12.5597 | | |
| | PreAct-ResNet-34 | $\mathbf{rate}_M$ | Time of Eligibility Track | 0.007 | 0.009 | 0.020 | | |
| | | | Time of Backward | 0.072 | 0.071 | 0.072 | | |
| | | | Time of both | 0.079 | 0.080 | 0.092 | | |
| | | | Memory Allocated [GB] | 5.4995 | 5.4982 | 5.4942 | | |
| | | $\mathrm{BPTT}_M$ | Time of Backward | 0.046 | 0.088 | 0.211 | | |
| | | | Memory Allocated [GB] | 3.7017 | 6.4778 | 11.969 | | |

ResNet-19, and VGG-11—and timesteps (T=2, 4, 6). The results demonstrate that rate-based back-propagation maintains competitive accuracy with BPTT across different architectures and timestep settings on benchmark datasets.

## D.3 Comprehensive Evaluation of Training Costs

To enhance the understanding of the scalability of the proposed method, we extended our analysis to include training costs across the CIFAR-100 and ImageNet datasets, utilizing additional network architectures as detailed in Table 6. This comprehensive evaluation aimed to assess the impact of varying time steps on performance, memory, and time costs. We integrated the computation of eligibility traces during the forward process, ensuring a fair comparison by incorporating these

iterative computations into the overall cost assessment. The results reveal that the total cost of rate-based backpropagation demonstrates a clear advantage over BPTT when timesteps $T \geq 2$, which underscores the efficiency of the proposed method approach in managing computational resources while maintaining comparative performance across various datasets and network architectures.

## E   Social Impacts and Limitations

There is no direct negative societal impact since this work centers on enhancing the training efficiency of SNNs. SNNs inherently require less energy for inference compared to ANNs, helping reduce carbon dioxide emissions. The methods developed in this work further optimize SNNs training by improving both memory and time efficiency, potentially reducing the overall resource consumption and environmental footprint of training processes. Regarding limitations, this work primarily compares with BPTT baselines, and there is potential for incorporating strategies from state-of-the-art techniques in future work. Moreover, the proposed method is tailored for tasks that utilize rate-coding, designed to efficiently capture spatial rate-based feature representations to enhance training; therefore, it necessitates further adaptation to effectively manage sequential tasks. Future efforts may need to delve deeper into adapting the dynamic characteristics of spikes and robustly designing training hyperparameters, ensuring compatibility with rate-based backpropagation and extending applicability to a wider range of applications.

