# OpenReview forum: "Advancing Training Efficiency of Deep Spiking Neural Networks through Rate-based Backpropagation"
_NeurIPS.cc/2024/Conference — NeurIPS 2024 poster_

### Official Review · Reviewer_Fw27 · 2024-06-21

**Soundness:** 3
**Presentation:** 3
**Contribution:** 3
**Rating:** 7
**Confidence:** 4

**Summary:**

Recent research indicates that rate-coding is crucial for information representation in deep Spiking Neural Networks (SNNs) trained via Backpropagation Through Time (BPTT). Building on this insight, a new training strategy called rate-based backpropagation has been developed to leverage rate-based representations, reducing the complexity of BPTT. This approach focuses on averaged dynamics to simplify the computational graph, thereby lowering memory and computational requirements. Theoretical and empirical analyses demonstrate that this method closely approximates BPTT's gradient optimization, maintaining comparable performance while surpassing other efficient training techniques. This advancement is poised to enable more scalable and resource-efficient SNN training, particularly in environments with limited resources.

**Strengths:**

1.	The paper is very well written and documented.
2.	The contributions have been discussed comprehensively.
3.	The experiments have been conducted on multiple benchmarks.

**Weaknesses:**

Some important details (such as the top-level algorithm of the proposed rate-based backpropagation method and details of the experimental setup) are reported in the appendix, while, due to their importance, they should be moved to the main manuscript.

**Questions:**

1.	Can the proposed rate-based backpropagation be implemented on existing neuromorphic chips with learning capabilities?
2.	Looking at the results in Fig.4, the impact of the number of timesteps on the time and memory looks constant. How have specific numbers of timesteps been selected for each dataset?

**Limitations:**

The limitations and societal impacts have been discussed in Appendix D.

---

> ### Author Rebuttal · Authors · 2024-08-07
>
> #### **W1: Some important details (such as the top-level algorithm of the proposed rate-based backpropagation method anddetails of the experimental setup) are reported in the appendix, while, due to their importance, they should be movedto the main manuscript.**
>
> Thank you for your suggestion. We appreciate your feedback and will consider moving more details to the main manuscript to facilitate a clearer understanding of the technical aspects for the readers.
>
> #### **Q1: Can the proposed rate-based backpropagation be implemented on existing neuromorphic chips with learning capabilities?**
>
> This is indeed an interesting point. The proposed rate-based backpropagation highlights the significant aspect of training efficiency, which is very pertinent. To our knowledge, some neuromorphic chips [1,2] are already incorporating online learning schemes that perform backpropagation at specific timesteps while maintaining eligibility traces at the neural level. We would say that architectures supporting online training could easily adapt to support our proposed method. Computationally, our approach offers an effective optimization of both computational and memory complexity compared to BPTT. Thus, on neuromorphic chips that allow for custom configurations, such as Tianjic [3] and Loihi [4], rate-based backpropagation would be more practical than BPTT in terms of computational cost, storage demands, and communication overhead.
>
>
> #### **Q2: Looking at the results in Fig.4, the impact of the number of timesteps on the time and memory looks constant. How have specific numbers of timesteps been selected for each dataset?**
>
> Thank you for your question. There are two aspects to consider in answering this. First and foremost, it's crucial to present our method as a fair comparison with recent works to determine whether the performance is compelling. Thus, we align the number of timesteps with those used in previous studies. Secondly, the number of timesteps used to train our SNN models indeed affects the inference latency, which is a key concern in the field of direct learning. Therefore, we often consider the tradeoff between higher inference accuracy and larger timesteps (which result in higher latency). The timesteps chosen for this study appear to be a good balance in benchmark scenarios.
>
>
> ## Reference
> [1] Frenkel, Charlotte, and Giacomo Indiveri. "ReckOn: A 28nm sub-mm2 task-agnostic spiking recurrent neural network processor enabling on-chip learning over second-long timescales." 2022 IEEE International Solid-State Circuits Conference (ISSCC). Vol. 65. IEEE, 2022.
>
> [2] Rostami, Amirhossein, et al. "E-prop on SpiNNaker 2: Exploring online learning in spiking RNNs on neuromorphic hardware." Frontiers in Neuroscience 16 (2022): 1018006.
>
> [3] Pei, Jing, et al. "Towards artificial general intelligence with hybrid Tianjic chip architecture." Nature 572.7767 (2019): 106-111.
>
> [4] Davies, Mike, et al. "Loihi: A neuromorphic manycore processor with on-chip learning." Ieee Micro 38.1 (2018): 82-99.

---

> > ### Comment · Reviewer_Fw27 · 2024-08-11
> > **Response to rebuttal**
> >
> > In light of the other reviews and the author's rebuttal, my score is confirmed.

---

### Official Review · Reviewer_B3hA · 2024-07-06

**Soundness:** 3
**Presentation:** 3
**Contribution:** 2
**Rating:** 6
**Confidence:** 4

**Summary:**

This paper presents a novel rate-based backpropagation method for spiking neural network (SNNS) training, which effectively separates the time-dependent backpropagation (BPTT) process and thus reduces computational and memory costs. The method employs a rate-encoded approximation to capture the basic information and is validated by empirical experiments on various datasets, demonstrating that it is superior in terms of training efficiency and accuracy when compared to the traditional BPTT.

**Strengths:**

1. Empirical results on multiple datasets (CIFAR-10, CIFAR-100, ImageNet, CIFAR10-DVS) support the theoretical claims and ensure accuracy while reducing memory and time costs.
2. The paper is well-written, clearly explaining the proposed method, theoretical underpinnings, and experimental validation.

**Weaknesses:**

1.	In lines 53-55, this paper mentions that the proposed method reduces training time, but there is no relevant experimental proof in the experiments section.

**Questions:**

1. In lines 223-234, the reference to 'when cosine similarity close to 1 is interpreted as a high degree of consistency in the direction of the variable', does it take into account the effects of data distribution and noise, which may also occur in the case of uneven data distribution. Can additional experiments or theories be added to rule out the effect of data distribution and noise on the hypothesis presented in lines 223-234?
2. The approach proposed in the paper seems to be very similar to the one described in reference [1]. Although the general direction of the two is different, the core idea seems to be the same. Could you please explain the difference between your approach and the one outlined in reference [1]?
3. In Section 5.3, in the experiments evaluating the effect of time step on accuracy and training, only one dataset, CIFAR-10, was used. Could the experiment be supplemented with experiments using other datasets to demonstrate the scalability of the proposed method for larger values of T?
4. In the caption of Fig. 3, is the placeholder '#' missing from T{timesteps}?

Reference:
[1] Bu, Tong, et al. "Rate gradient approximation attack threats deep spiking neural networks." Proceedings of the IEEE/CVF Conference on Computer Vision and Pattern Recognition. 2023.

**Limitations:**

The authors fully explain the limitations and potential social implications of their work.

---

> ### Author Rebuttal · Authors · 2024-08-07
>
> #### **W1: In lines 53-55, this paper mentions that the proposed method reduces training time, but there is no relevant experimental proof in the experiments section.**
> Thank you for your suggestion. We have added more experiments on training costs to strengthen the experimental proof of training efficiency, as shown in the global response. These results will be included in the paper. Thank you for pointing this out.
>
> #### **Q1: In lines 223-234, the reference to 'when cosine similarity close to 1 is interpreted as a high degree of consistency in the direction of the variable', does it take into account the effects of data distribution and noise, which may also occur in the case of uneven data distribution. Can additional experiments or theories be added to rule out the effect of data distribution and noise on the hypothesis presented in lines 223-234?**
>
>
> Thank you for your suggestions.
> First, we have supplemented empirical experiments on DVS-CIFAR10 as a validation in the case of uneven data distribution (please see Figure in the global response). The observations confirm that, even in data with a degree of temporal information, the empirical validation of the assumptions remains consistent with expectations.
> Furthermore, we would like to clarify Theorem 2 mentioned in the paper. We derived a general approximation error in Theorem 2, which provides a proof concerning the error bounds. This can demonstrate that the approximation errors caused by non-ideality of assumptions are well bounded. We believe that both the effects of uneven data distribution and noise can be generally considered as factors causing approximation errors in our assumptions, and Theorem 2 has already accounted for these scenarios.
>
>
> #### **Q2: The approach proposed in the paper seems to be very similar to the one described in reference [1]. Although the general direction of the two is different, the core idea seems to be the same. Could you please explain the difference between your approach and the one outlined in reference [1]?**
>
> Here are the main differences between our work and reference [1]:
>
> 1. Our method to gradient computation differ significantly. We likely to conclude the concepts into, gradient-computing via stochastic dynamic process v.s. deterministic closed-form. [1] derives gradient relationships based on a deterministic relationship assumption between inputs and outputs in a closed-form manner, similar to the methodologies used for ANN-to-SNN conversions [2]. Conversely, our method adopts a stochastic dynamic process for gradient computation, which follows the ideas of decoupling BPTT-based direct-learning methods [3,4].
> 2. The application domains of the two methods vary, influencing dinstinct objectives. The method in [1] is used primarily in adversarial attack scenarios, focusing on the input data of the entire model. In such cases, the requirement for strict adherence to the exact gradient computation process is more relaxed, and deviations are somewhat permissible. In contrast, our method is designed for training deep SNNs, necessitating more precise gradient computations for all weights. Our approach supports more strict gradients within rate-based backpropagation, and experiments have demonstrated the proposed method can achieve gradients akin to BPTT.
>
> We will include a topic in the appendix to provide a more detailed description of how our work differs from related work. Thank you for your comment.
>
>
> #### **Q3: In Section 5.3, in the experiments evaluating the effect of time step on accuracy and training, only one dataset,CIFAR-10, was used. Could the experiment be supplemented with experiments using other datasets to demonstrate the scalability of the proposed method for larger values of T?**
>
> Thank you for your suggestion. Our additional experiments evaluated the effects of timestep on accuracy, memory, and time costs across the CIFAR-100 and ImageNet datasets using various network architectures (in global PDF). These results are consistent with the discussions in our manuscript and demonstrate the scalability of the proposed method. We will include the additional results into the updated version.
>
> #### **Q4: In the caption of Fig. 3, is the placeholder '#' missing from T{timesteps}?**
>
> Thank you for pointing that out; it will be corrected.
>
>
> ## Reference
> [1] Bu, Tong, et al. "Rate gradient approximation attack threats deep spiking neural networks." Proceedings of the IEEE/CVF Conference on Computer Vision and Pattern Recognition. 2023.
>
> [2] Bu, Tong, et al. "Optimal ANN-SNN Conversion for High-accuracy and Ultra-low-latency Spiking Neural Networks." International Conference on Learning Representations.
>
> [3] Bellec, Guillaume, et al. "A solution to the learning dilemma for recurrent networks of spiking neurons." Nature communications 11.1 (2020): 3625.
>
> [4] Xiao, Mingqing, et al. "Online training through time for spiking neural networks." Advances in neural information processing systems 35 (2022): 20717-20730.

---

### Official Review · Reviewer_YGtz · 2024-07-07

**Soundness:** 3
**Presentation:** 2
**Contribution:** 3
**Rating:** 5
**Confidence:** 4

**Summary:**

This work falls into the category of efficient SNN training methods. This paper proposes a reduced computational graph to reduce the memory and computational demands of SNNs training. This work has the potential to train SNNs on resource-limited devices. The paper evaluates the methods on CIFAR-10, CIFAR-100, ImageNet, and other datasets.

**Strengths:**

This paper addresses the issue of high time and memory costs in training spiking neural networks.

This paper provides solid theoretical insights into the error bound and its relation to SNN BPTT training.

The results of this work are comparable to the performance of the BPTT counterpart.

**Weaknesses:**

Not a clear comparison of the differences with existing e-prop methods in terms of methodology.

No generalization results on hyperparameters (e.g., $\lambda$) are presented in this work. I raise this question because most work on SNNs uses large values of $\lambda$, but this work used 0.2 as $\lambda$.

**Questions:**

Why did the authors approximate the spiking rate directly with the mean over timesteps, instead of using a running mean with a decay parameter $\lambda$, which would more closely approximate the rate in the leaky integration mode?

In Line 151, page 5, what does 𝑑 represent in $\frac{\partial I}{partial c} = Id$?

Please elaborate further on the differences between rateM and rateS. The authors state that 'rateM represents the multi-step training mode where T loops are embedded within layers, while rateS refers to the single-step training mode with T loops outside the layer.'

Regarding robustness to $\lambda$ In the paper, the neuronal parameter $\lambda$ is set to 0.2. Can you provide experiments with other values of $\lambda$, such as 0.4, 0.6, 0.8, and 1.0?

I believe that the training cost in Fig. 4 should encompass not only the backward process but also the forward iteration process (which also contributes to the cost).

**Limitations:**

See weakness and questions.

---

> ### Author Rebuttal · Authors · 2024-08-07
>
> #### **W1: Not a clear comparison of the differences with existing e-prop methods in terms of methodology.**
>
> Thank you for your comments. In our paper, we compare our method with online-learning akin to e-prop [1], as OTTT[2,3,4], SLTT[3], and OS[4], with results shown in Table 1. Descriptions of online methods can be found in Lines 29-32, 81-87, 161-166, and 251-255.
>
> The core differences in implementation with these methods, as highlighted in Fig. 1b and 1c, are that:
> 1. Unlike online methods that require spatial backpropagation at every timestep, our proposed method conducts this process only once at the final timestep, effectively compressing the multiple instances of spatial backpropagation into a single occurrence.
> 2. Methodologically, the proposed rate-based method differs from e-prop in that we do not focus on specific temporal dynamics among all time slots but rather aim to efficiently train networks by capturing dominant rate-based features.
>
> Thank you for pointing this out. We will ensure these details are more clearly articulated in the paper.
>
> #### **W2: No generalization results on hyperparameters (e.g. $\lambda$) are presented. Most work on SNNs uses large values of $\lambda$, but this work used 0.2.**
>
> Thank you for your comment. We configured our training environment and hyperparameters based on setups from existing works on direct training-based deep SNNs [5,6,7,8,2,3,9,4,10,11]. In these works, the decay parameter $\lambda$ is set at 0.5 in [2,4,11], 0.25 in [8], 0.2 in [7,9,10], and 0.09 in [3]. It is observed that direct training methods often employ $\lambda$ of 0.5 or lower. We chose 0.2 to align with training settings used in [7,10].
>
> #### **Q1: Why did the authors approximate the spiking rate directly with the mean over timesteps, instead of using a runningmean with a decay parameter, which would more closely approximate rate in the leaky integration mode?**
>
> Thank you for your comments.
>
> Firstly, the choice of averaging over timesteps for measuring the spiking rate aligns better with our method:
> 1. Using a running mean with a decay parameter [12,3], which measures a scaled running average of the rate, typically retains temporal dimensions where weights for spikes closer in time are higher. This approach often requires constructing rate-based backpropagation separately for different time slots.
> 2. In contrast, our approach involves constructing a single rate-based chain rule for spatial-only backward propagation at the final timestep. Therefore, approximating the mean rate over the entire neural dynamics is more suitable for our goals.
>
> Regarding the second question, considering the rate estimation at time $T$, our method gives $r_1=\frac{\sum_t s_t}{T}$, while a running mean with a decay parameter would give $r_2=\frac{\sum_t \lambda^{(T-t)}s_t}{\sum_t \lambda^{(T-t)}}$. If the spike sequence follows frequency coding with a spike rate $r$ (assuming $s_t \sim Bernoulli(r)$), then $\mathbb{E}[r_1]=\frac{\sum_t \mathbb{E[s_t]}}{T}=r$ and $\mathbb{E}[r_2]=\frac{\sum_t \lambda^{(T-t)}\mathbb{E}[s_t]}{\sum_t \lambda^{(T-t)}}=r$. This supports that both methods provide unbiased estimates of the rate.
>
> In summary, our choice of averaging over timesteps for rate approximation is driven by its compatibility with our single-step spatial-only backpropagation approach, simplifying the implementation and aligning with our method's objectives effectively.
>
>
> #### **Q2: In Line 151, page 5, what does $d$ represent?**
>
> We apologize for the ambiguity; $Id$ represents the identity matrix. We will add clarification.
>
>
> #### **Q3: Please elaborate further on the differences between rateM and rateS.**
>
> $rate_M$ and $rate_S$ cater to multi-step and single-step training modes, respectively, each adapting batch normalization differently as detailed in the the Appendix B.2 and B.3. We support both modes to demonstrate method’s flexibility and to facilitate fair comparisons with related methods.
>
>
>
> #### **Q4: Can you provide experiments withother values of $\lambda$, such as 0.4, 0.6, 0.8, and 1.0?**
> Additional experiments on CIFAR10 with T=6 setting using ResNet-18
>
> |$\lambda$|0.1|0.2|0.4|0.5|0.6|0.8|1.0|
> |:-:|:-:|:-:|:-:|:-:|:-:|:-:|:-:|
> |$rate_M$|95.45|95.8|95.65|95.34|95.11|93.82|92.79|
> |BPTT|95.51|95.69|95.48|95.19|95.31|94.65|94.7|
>
> The results indicate that when $\lambda$ is within the range of $[0.1, 0.5]$, the performance of our rate-based method aligns closely with that of BPTT. However, as it increases into $[0.8, 1.0]$, the performance of both BPTT and rate-based methods declines, with a more pronounced decrease in the rate-based method. We believe the primary reason for this phenomenon is the non-ideality in the assumption of independent distributions used in Theorem 1, $\mathbb{E}[\delta^{(s^l)}_t \kappa^l_t ] = \mathbb{E}[\delta^{(s^l)}_t]\mathbb{E}[\kappa^l_t]$, which leads to larger approximation errors at higher $\lambda$ values, as discussed in Theorem 2 where $\epsilon$ is amplified, thus magnifying the gradient difference with BPTT. We will address this issue in the appendix, discussing the implications and acknowledging it as a limitation in our study.
>
>
> #### **Q5: I believe that the training cost in Fig. 4 should encompass not only the backward process but also the forward iteration process.**
>
> Thank you for your feedback. We have supplemented our work with more detailed experiments on training costs, including the extra costs incurred during the forward iteration process as shown in the global response. Firstly, the total time cost combined with the computation of eligibility traces in forward, still shows a clear advantage over BPTT when $T\geq2$. Additionally, it is observed that the computation of eligibility traces is less intensive than the backward process when $T\leq 8$.
>
> We will include the supplemented experimental results into the paper and acknowledge that the computation of eligibility traces during the forward process requires further optimization.

---

> ### Author Response · Authors · 2024-08-07
>
> ## Reference
>
> [1] Bellec, Guillaume, et al. "A solution to the learning dilemma for recurrent networks of spiking neurons." Nature communications 11.1 (2020): 3625.
>
> [2] Xiao, Mingqing, et al. "Online training through time for spiking neural networks." Advances in neural information processing systems 35 (2022): 20717-20730.
>
> [3] Meng, Qingyan, et al. "Towards memory-and time-efficient backpropagation for training spiking neural networks." Proceedings of the IEEE/CVF International Conference on Computer Vision. 2023.
>
> [4] Zhu, Yaoyu, et al. "Online stabilization of spiking neural networks." The Twelfth International Conference on Learning Representations. 2024.
>
> [5] Fang, Wei, et al. "Spikingjelly: An open-source machine learning infrastructure platform for spike-based intelligence." Science Advances 9.40 (2023): eadi1480.
>
> [6] Guo, Yufei, Xuhui Huang, and Zhe Ma. "Direct learning-based deep spiking neural networks: a review." Frontiers in Neuroscience 17 (2023): 1209795.
>
> [7] Wu, Yujie, et al. "Spatio-temporal backpropagation for training high-performance spiking neural networks." Frontiers in neuroscience 12 (2018): 331.
>
> [8] Zheng, Hanle, et al. "Going deeper with directly-trained larger spiking neural networks." Proceedings of the AAAI conference on artificial intelligence. Vol. 35. No. 12. 2021.
>
> [9] Guo, Yufei, et al. "Recdis-snn: Rectifying membrane potential distribution for directly training spiking neural networks." Proceedings of the IEEE/CVF conference on computer vision and pattern recognition. 2022.
>
> [10] Wang, Ziming, et al. "Adaptive smoothing gradient learning for spiking neural networks." International Conference on Machine Learning. PMLR, 2023.
>
> [11] Deng, Shikuang, et al. "Surrogate module learning: Reduce the gradient error accumulation in training spiking neural networks." International Conference on Machine Learning. PMLR, 2023.
>
> [12] Meng, Qingyan, et al. "Training high-performance low-latency spiking neural networks by differentiation on spike representation." Proceedings of the IEEE/CVF conference on computer vision and pattern recognition. 2022.

---

### Official Review · Reviewer_TjVW · 2024-07-10

**Soundness:** 3
**Presentation:** 3
**Contribution:** 2
**Rating:** 6
**Confidence:** 5

**Summary:**

This paper proposes a rate-based SNN training method, which can effectively reduce memory and time cost during training. They proved the efficiency of the rate-based back-propagation training and demonstrate that the rate-based training outperforms other back-propagation methods.

**Strengths:**

The rate-based method achieves better performance and uses less computing resource compared with BPTT, which is impressive.

This paper is well-written and well-organized.

**Weaknesses:**

The novelty is weak. There are two previous works that share similar idea with this paper, since they all use rate-based backpropagation [1,2]. The author needs to briefly explain the differences between these papers.

The rate-based backpropagation is not suitable for sequential tasks.

[1] Li, Yuhang, et al. "Differentiable spike: Rethinking gradient-descent for training spiking neural networks." Advances in Neural Information Processing Systems (2021).
[2] Bu, Tong, et al. "Rate gradient approximation attack threats deep spiking neural networks." Computer Vision and Pattern Recognition (2023).

**Questions:**

The authors introduce the rate-coding approximation forward propagation in Section 4.1. Is this forward propagation method also used during inference?

What is the performance of rate$_s$ on ImageNet dataset?

**Limitations:**

See in weaknesses.

---

> ### Author Rebuttal · Authors · 2024-08-07
>
> #### **W1: The novelty is weak. There are two previous works that share similar idea with this paper, since they all use rate-basedbackpropagation [1,2]. The author needs to briefly explain the differences between these papers.**
>
>
> Thank you for your comments, which have prompted further clarification of our work's novelty and contributions.
>
> Firstly, regarding the differentiation from references [1] and [2]:
>
> 1. The work in [1] focuses primarily on the calibration and modification of STE [3] (or "surrogate-gradients" commonly used in SNNs). In contrast, our approach significantly simplifies the chain rule application by compressing backpropagation within the spatial dimensions to enhance training efficiency.
> 2. Meanwhile, [2] derives gradient relationships based on a deterministic relationship assumption between inputs and outputs in a closed-form manner, similar to the methodologies used for ANN-to-SNN conversions [4]. Conversely, our method adopts a stochastic dynamic process for gradient computation, which follows the ideas of decoupling BPTT-based direct-learning methods. Additionally, while [2] targets the adversarial attack domain where precise gradient calculations are not stringent, our proposed method as a training method requires more accurate gradient computations, demonstrated experimentally to achieve gradients akin to those of BPTT.
>
> Furthermore, other works bind ANN and SNN training together in what's known as tandem training [5]. However, these approaches cannot connect well to established BPTT, limiting their accuracy and competitiveness with BPTT benchmarks. There are also efforts considering rate-based gradient graphs [6], aimed at optimizing training costs, but these are similarly limited by closed-form derivations like [4], unable to bypass the coupling of batch normalization layers for purely spatial backpropagation.
>
> We appreciate your suggestion for greater clarity, and we will include a detailed discussion in the appendix to better delineate how our method differs from other rate-based approaches. Thank you for prompting this improvement.
>
> To further clarify our novelty and contribution, it is important to note that our work's originality does not claim to be the first to propose rate-based backpropagation. Rather, our significant contribution lies in being the first to effectively implement the concept of rate-based backpropagation as a training efficiency strategy, demonstrating its potential to perform comparably with BPTT. This is not intended as a perfect substitute for BPTT but as an effective complement and enhancement, forming a promising combined approach. We believe this technical solution, which optimizes the training process of deep SNNs, holds potential developmental benefits for the community.
>
>
> #### **W2: The rate-based backpropagation is not suitable for sequential tasks.**
> We acknowledge that rate-based approaches [2,4,5,6] including our proposed method, indeed face challenges in handling sequential tasks, which is a recognized limitation of such strategies. This work does not aim to address the difficulties of applying rate-based methods to sequential tasks. Instead, our focus is primarily on tasks that rely mainly on rate-coding, and the proposed method is specifically designed to capture the rate-based feature representation of deep SNNs to enhance training efficiency. Thank you for highlighting this point; we will include it in the limitations section.
>
>
> #### __Q1: The authors introduce the rate-coding approximation forward propagation in Section 4.1. Is this forward propagation method also used during inference?__
>
> No, the rate-coding approximation is purely used during the training phase and does not participate in the inference process. The inference strictly adheres to the conventional protocols of deep SNNs to ensure fair comparisons.
>
> #### __Q2: What is the performance of $rate_S$ on ImageNet dataset?__
>
> Additional experiments have been conducted to evaluate the performance of $rate_S$ on the ImageNet:
>
> ||Timesteps | Top-1 Acc (%)|
> |:-:|:-:|:-:|
> |SEW-ResNet-34|4|65.656         |
> |PreAct-ResNet-34|4|69.578       |
>
> The results confirm that the $rate_S$ method is effective on ImageNet. We will integrate these results into Table 1.
>
>
>
> ## Reference:
> [1] Li, Yuhang, et al. "Differentiable spike: Rethinking gradient-descent for training spiking neural networks." Advances in Neural Information Processing Systems 34 (2021): 23426-23439.
>
> [2] Bu, Tong, et al. "Rate gradient approximation attack threats deep spiking neural networks." Proceedings of the IEEE/CVF Conference on Computer Vision and Pattern Recognition. 2023.
>
> [3] Bengio, Yoshua, Nicholas Léonard, and Aaron Courville. "Estimating or propagating gradients through stochastic neurons for conditional computation." arXiv preprint arXiv:1308.3432 (2013).
>
> [4] Bu, Tong, et al. "Optimal ANN-SNN Conversion for High-accuracy and Ultra-low-latency Spiking Neural Networks." International Conference on Learning Representations.
>
> [5] Wu, Jibin, et al. "A tandem learning rule for effective training and rapid inference of deep spiking neural networks." IEEE Transactions on Neural Networks and Learning Systems 34.1 (2021): 446-460.
>
> [6] Meng, Qingyan, et al. "Training high-performance low-latency spiking neural networks by differentiation on spike representation." Proceedings of the IEEE/CVF conference on computer vision and pattern recognition. 2022.

---

> > ### Comment · Reviewer_TjVW · 2024-08-10
> > **Response for the Rebuttal**
> >
> > In the rebuttal, the authors distinguished between this rate-based BP and the previous works[1,2], which I found convincing. Since they addressed my major concerns, I will raise my score.

---

### Author Rebuttal · Authors · 2024-08-07

Thank you to all the reviewers for your constructive comments and suggestions. We have addressed all the weaknesses and questions raised by the reviewers; detailed responses can be found in the corresponding sections of the rebuttal for each reviewer. In response to the reviewers' feedback, we have conducted additional experiments, including empirical validation on DVS-CIFAR10 and further experimental validation on performance and training costs, the results of those two are included in the global PDF.

---

### Decision · Program_Chairs · 2024-09-25

**Decision:**

Accept (poster)

**Comment:**

This paper explores improved techniques for training spiking neural networks using backpropagation-through-time (BPTT). Specifically, the authors develop a technique relying on averaged rates-of-fire rather than detailed spiking dynamics. The authors show that this approach streamlines the computational graph, leading to less memory and time required for training. They demonstrate using various image categorization benchmarks that their approach leads to performance that is comparable to BPTT and better than existing efficient approaches for spiking networks.

The reviews for this paper were generally positive. Reviewers raised some concerns about novelty and clarity, which the authors sought to address in their rebuttal. Following this, the final average score was 6.0, and there was no reviewer that felt that this paper should be rejected. The AC concurred, and so, a decision to accept as a poster was reached.